# Regionally refined testbed in E3SM Atmosphere Model Version 1 (EAMv1) and applications for high-resolution modelling

Qi Tang[1], Stephen A. Klein[1], Shaocheng Xie[1], Wuyin Lin[2], Jean-Christophe Golaz[1], Erika L. Roesler[3], Mark A. Taylor[3], Philip J. Rasch[4], David C. Bader[1], Larry K. Berg[4], Peter Caldwell[1], Scott E. Giangrande[2], Richard Neale[5], Yun Qian[4], Laura D. Riihimaki[4], Charles S. Zender[6], Yuying Zhang[1], and Xue Zheng[1]

[1]Lawrence Livermore National Laboratory, Livermore, CA 94550, USA
[2]Brookhaven National Laboratory, Upton, NY 11973, USA
[3]Sandia National Laboratory, Albuquerque, NM 87185, USA
[4]Pacific Northwest National Laboratory, Richland, WA 99352, USA
[5]National Center for Atmospheric Research, Boulder, CO 80305, USA.
[6]Departments of Earth System Science and Computer Science, University of California, Irvine, Irvine, CA 92697, USA

*Correspondence to*: Qi Tang (tang30@llnl.gov)

**Abstract.** Climate simulations with more accurate process-level representation at finer resolutions (< 100 km) is a pressing need in order to provide more detailed actionable information to policy-makers regarding extreme events in a changing climate. Computational limitation is a major obstacle for building, and running high-resolution (HR, here 0.25º average grid spacing at the equator) models (HRM). A more affordable path to HRM is to use a global regionally refined model (RRM), which only simulates a portion of the globe at HR while the remaining is at low-resolution (LR, 1º). In this study, we compare the Energy Exascale Earth System Model (E3SM) atmosphere model version 1 (EAMv1) RRM with the HR mesh over the contiguous United States (CONUS) to its corresponding globally uniform LR and HR configurations, as well as to observations and reanalysis data. The RRM has a significantly reduced computational cost (roughly proportional to the HR mesh size) relative to the globally uniform HRM. Over the CONUS, we evaluate the simulation of important dynamical and physical quantities as well as various precipitation measures. Differences between the RRM and HRM over the HR region are predominantly small, demonstrating that the RRM reproduces the precipitation metrics of the HRM over the CONUS. Further analysis based on RRM simulations with the LR vs. HR model parameters reveals that RRM performance is greatly influenced by the different parameter choices used in the LR and HR EAMv1. This is a result of the poor scale-aware behaviour of physical parameterizations, especially for variables influencing sub-grid scale physical processes. RRM can serve as a useful framework to test physics schemes across a range of scales, leading to improved consistency in future E3SM versions. Applying nudging-to-observations techniques within the RRM framework also demonstrates significant advantages over a free-running configuration for use as a testbed, and as such represents an efficient and more robust physics testbed capability. Our results provide additional confirmatory evidence that the RRM is an efficient and effective testbed for HRM development.

# 1 Introduction

A key goal of the United States (US) Department of Energy (DOE) Energy Exascale Earth System Model (E3SM) project (formally known as the Accelerated Climate Modeling for Energy (ACME)) is to develop a high-resolution (HR, 0.25º or finer in the horizontal) fully-coupled Earth system model for climate simulation and prediction (Bader et al., 2014). Testing new physical parameterizations and tuning loosely constrained parameters within existing parameterizations are important steps of model development. However, the computational cost of running a globally uniform HR model (HRM) is high. For example, a one-year 0.25º HR E3SM Atmosphere Model Version 1 (EAMv1) simulation requires 0.6 million core-hours on 675 "Knights Landing" (KNL, Intel Xeon Phi Processor 7250) nodes of the Cori supercomputer at the National Energy Research Scientific Computing Center (NERSC). A regionally refined model (RRM) capability (Ringler et al., 2008; Zarzycki and Jablonowski, 2014; Roesler et al., 2018), which only simulates a fraction of the globe at HR, is adopted by EAMv1 to reduce the computational cost of HR simulations and to examine the parameterization sensitivity to HR scales. The RRM simulation cost is usually dominated by the computational cost of the HR region, and thus the total model cost is roughly proportional to the size of the region with finer resolution, referred to as a "mesh" (typically chosen to be about 10% of the globe, making the cost about 10% of a uniform HRM simulation). In the ongoing E3SM phase II project, the RRM configuration is planned as a central tool to achieve the E3SMv2 science goal of understanding the relative impacts of global forcing versus regional influences of human activities on flood and drought in North America. RRM will be routinely used over North America to address DOE's goal to understand the Earth system changes affecting US energy-sector decisions. It will be also applied as a physics testbed to improve the scale-awareness of parameterizations in upcoming versions of E3SMv2 and v3 as well as an important strategy to perform larger ensemble of HR simulations. RRM is also a vital capability for progress towards an eventual global cloud-resolving model with 3 km horizontal grid spacing targeting E3SMv4 and beyond.

The RRM approach has been established and validated with other models over many regions of interest. For instance, Zarzycki et al. (2014) showed the effectiveness of an RRM with aquaplanet experiments using the Community Atmosphere Model (CAM). Zarzycki and Jablonowski (2014, 2015) demonstrated improved skill in simulating tropical cyclones in CAM with a refined mesh over the North Atlantic. Rhoades et al. (2016) and Wu et al. (2017) depicted that the variable-resolution (VR) Community Earth System Model (CESM) was able to accurately capture the climatology and seasonality of important variables over mountain regions. Huang and Ullrich (2017) reproduced the geographic patterns of historical precipitation climatology over the western US with the VR-CESM. Gettelman et al. (2018) performed comprehensive tests of a VR dynamical core in CESM2 and showed that VR grids were feasible alternative to conventional nesting for regional climate research. Roesler et al. (2018) found that refining the grid over the contiguous United States (CONUS) did not exert a noticeable influence on the global circulation in the EAM version 0 (EAMv0, which is almost identical to CAM5.3 except

for some minor tunings and bug fixes). These earlier studies have demonstrated that RRM can be used as an effective tool to study important climate features over regions of interest with high resolution.

Compared to EAMv0, EAMv1 (Rasch et al., 2019) includes significant changes to its physics, substantially increased
vertical resolution, retuning, and bug fixes (Zhang et al., 2018). All these changes cause the model to behave very differently from EAMv0, especially in terms of regional clouds and precipitation characteristics (Xie et al., 2018). Given these substantial model changes and the critical role that RRM will play in future E3SM scientific applications, this paper documents further scientific analysis of RRM behaviour with EAMv1. We contrast simulations between the RRM and the globally uniform HR EAMv1 over the RRM region, with the goal to provide more insights into the EAMv1 RRM capability
to the user community. This study emphasizes hydrology-related simulation skill over North America: a key element of the E3SM Water Cycle science driver. We investigate whether RRM reproduces the same performance as HRM of these fields enabling it to be used as an effective physics testbed for understanding physical processes and improving their representations in EAMv1 and in future versions. In addition, EAMv1 physical parameterizations (and in particular the cloud parameterizations) are not inherently scale-aware and hence require retuning when increasing model horizontal or vertical
resolution. Unfortunately, this leads to two different parameter settings for EAMv1 high- and low-resolution models. It is key to determine how the two different parameter settings influence RRM performance, since most earlier studies just used the established low-resolution model parameters over the RRM domain, which may not yield optimal RRM results due to scale-aware shortcomings of the existing physical schemes.

This study centres mainly on "proof-of-concept" examples. More in-depth analysis of RRM behaviour will be reported in separate studies when RRM is more routinely used in E3SM phase II and by general users. In many EAMv1 application scenarios, it is expected that the RRM will be more feasible and practical than the HRM. This could include evaluation against regional measurements, uncertainty quantification studies that typically demand a large ensemble size (Qian et al., 2016, 2018), and users with limited computational resource. Findings from this study regarding the strengths and weaknesses
of the EAMv1 RRM configuration should provide valuable guidance for future RRM applications in the HR E3SM development and broad community use of the E3SM RRM.

Additionally, we provide detailed information on how to utilize the RRM capability with nudging for process-level understanding of model deficiencies. Similar to the hindcast approach (Phillips et al., 2004; Ma et al., 2015) used in climate
model evaluation, the nudging approach is able to maintain the large-scale dynamical state close to an observed state, and hence provides a better assessment of atmospheric physics performance. This nudged approach is particularly useful for those processes that are related to fast physics. Xie et al. (2012) and Ma et al. (2014) demonstrated a strong correspondence between short (a few days) and long (seasonal to annual) timescale systematic errors in climate models for fields related to fast physics, such as clouds and precipitation.

This paper is organized as follows. Section 2 provides an overview of the RRM EAMv1 and summarizes the setup of simulations and the observational datasets used for model evaluation. Results are shown in Section 3, including model climatologies over the CONUS domain where our RRM has its fine-resolution mesh, the analysis of quantities related to hydrological cycle, and an in-depth analysis of precipitation characteristics – the large-scale/convective partitioning, the intensity distribution, and the summertime diurnal cycle. Section 4 describes an example of running the nudged RRM. Section 5 provides a summary of this work and prospects for future studies.

## 2 Methodology

### 2.1 Model overview and experiment design

The E3SM project aims to build a global HR fully-coupled Earth system model for climate simulation and prediction on current and next-generation supercomputing facilities (Bader et al., 2014). Since all the simulations analysed here are atmosphere-only ones, we only provide information about the atmosphere model. Details about the coupled E3SM model can be found in (Golaz et al., 2019). EAMv1 originated from CAM5.3 (Neale et al., 2012), but has undergone substantial development. An overview of EAMv1 is given by Rasch et al. (2019). More details on the simulated cloud and precipitation characteristics and overview of the low- and high-resolution model tunings are provided in Xie et al. (2018). EAMv1 uses the spectral element dynamical core (Taylor and Fournier, 2010; Dennis et al., 2012) on a cubed-sphere computation grid with an explicit Runge-Kutta time integration scheme. This dynamical core has sustained scalability with increasing number of elements and processors (Fournier et al., 2004). Major changes in EAMv1 compared to its earlier version include substantially increased vertical resolution (72 vs. 30 vertical layers), a higher (~0.1 hPa compared to 2 hPa) model top, and improved physical parameterizations including the Cloud Layers Unified By Binormals (CLUBB) scheme (Golaz et al., 2002; Bogenschutz et al., 2013), updated cloud microphysics (MG2) (Gettelman and Morrison, 2015), predicted aerosols (the Modal Aerosol Module (MAM4)) (Liu et al., 2016), and a linearized ozone chemistry (Linoz2) (Hsu and Prather, 2009). Impacts of the new cloud physics and the increase in vertical resolution on EAMv1 simulated climate are documented in Xie et al. (2018) and Qian et al. (2018). In the present paper, we focus on the EAMv1 regionally refined testbed capability over the CONUS domain.

The CONUS regionally refined grids consist of LR and HR regions and a transition area between them (see Fig. 1a). The HR grid is located in the CONUS area. We created the regionally refined grid with the offline software tool, Spherical Quadrilateral Grid Generator (SQuadGen, https://github.com/ClimateGlobalChange/squadgen). The effective resolutions for the LR and HR regions are 1° and 0.25°, respectively. Because of the horizontal resolution differences in the LRM, the HRM, and the RRM, the topography is represented differently in these configurations. We used a new tensor hyperviscosity formulation (Guba et al., 2014) to eliminate numerical noise and oscillations. Additional details about the CONUS RRM as

well as the topography data are reported in Roesler et al. (2018). It is worth mentioning that the RRM grids have also being generated and tested over the Tropical Western Pacific (TWP) and the Eastern North Atlantic (ENA).

In the present study, we mainly analyse the atmosphere-only simulations (see Table 1 and A1) forced by observed present-day climatologies of aerosol emissions, greenhouse gases, sea surface temperatures (SSTs) and sea ice concentrations. The simulations use an interactive E3SM land model on the same grids as the atmosphere. We run the EAMv1 with globally-uniform LR and HR grids as well as the CONUS RRM grid. All simulations are performed with the 72 vertical layers. Since the EAMv1 parameterizations are not scale-aware, both dynamical and physical parameters are adjusted to optimize the model performance at different resolutions (Xie et al., 2018). This leads to different parameter settings for the EAMv1 LRM and HRM. As shown in Table A1 of Xie et al. (2018), the differences are mainly in parameters that control convection and cloud microphysics. Thus, differences between LRM and HRM analysed in the following sections arise from different horizontal resolutions and parameter settings, as well as the different physics time steps. The LRM and HRM physics time steps are 30 minutes and 15 minutes, respectively. The dynamics use 3 layers of substepping. For the LRM (HRM), the Lagrangian vertical discretization timestep is 15 minutes (2.5 minutes), the horizontal discretization timestep is 5 minutes (75 seconds), and the explicit numerical diffusion timestep is 100 seconds (18.75 seconds). The RRM uses the same dynamics time steps over the LR and HR domains. For the purpose of mimicking the HRM behaviours, we opt to use the same dynamical and physical parameters and time step for the RRM control simulation as in the HRM. Besides the RRM control case, we also perform an RRM test (RRM_LR) with the LRM dynamical and physical parameters. Comparing these two RRM results, we are able to explore the impact of different parameter settings on the RRM performance, which is not possible for conventional RRM studies with only the LR parameters.

Current climate models commonly suffer from systematic biases in simulating climate mean states of clouds and precipitation associated with flaws in physical parameterizations (e.g., Klein et al., 2013; Ma et al., 2014). However, compensating errors from nonlinear feedback mechanisms also contribute to climate mean biases, making it a challenge to pin the errors to specific parameterizations. The numerical weather prediction technique (Phillips et al., 2004), also known as the transpose-AMIP (Williamson, 2013) or hindcast (Ma et al., 2015) approach, has been increasingly used in climate models, including EAMv1, to understand and reduce model errors associated with fast atmospheric physical processes. Similar to the hindcast method, the EAMv1 RRM can be run in a nudging configuration to diagnose parameterization-related errors and helping to guide development in HR E3SM. More guidance on using the nudged RRM approach will be discussed in Section 4.

To demonstrate the nudging capability, we perform an RRM simulation nudged to the European Centre for Medium-range Weather Forecasting Interim (ERAI) analysis fields of horizontal velocities (U and V) (Dee et al., 2011) with a 6-hour relaxation time scale. The nudged simulation uses the prescribed weekly, 1° spatial resolution SSTs and sea ice from the

National Oceanic and Atmospheric Administration Optimum Interpolation analysis data (Reynolds et al., 2002). In addition, we conduct a LR atmosphere-only simulation with time-evolving forcings (i.e., AMIP-style) to compare with the nudging simulation. Output from the Cloud Feedback Model Intercomparison Project Observation Simulator Package (COSP) (Bodas-Salcedo et al., 2011) is used to compare with cloud observations from satellites (Zhang et al., 2019). All free-running

simulations are run for a period of 5 years. The first year is considered as spin-up, thus we study the results from the last 4 years. The nudging run simulates year 2011, whereas the AMIP results are extracted for year 2011 from a long simulation starting from 1870 (Golaz et al., 2019). Model output is stored as monthly and hourly averages.

## 2.2 Evaluation datasets

Skilful depictions of the large-scale circulation and sub-grid scale physics are essential for more realistic model simulations

of the atmospheric hydrological cycle. We choose evaluation variables to cover both aspects. Evaluation datasets are summarized in Table 2. Meteorological fields, such as geopotential height, surface pressure, winds, temperature, relative humidity, and precipitable water are from the ERAI reanalysis product (Dee et al., 2011). Seasonal precipitation climatology estimations are based on the Global Precipitation Climatology Project (GPCP) (Huffman et al., 2009). Daily precipitation observations are taken from the GPCP one-degree daily (1DD) data (Huffman et al., 2001). Hourly precipitation is compared

with the dataset collected by the Next-Generation Radar (NEXRAD) network (NOAA, 2013) and developed under the Climate Science for a Sustainable Energy Future (CSSEF) project (Zhang et al., 2005, 2011; Giangrande et al., 2014). Simulated cloud amount is verified against International Satellite Cloud Climatology Project (ISCCP) data (Rossow and Schiffer, 1991). In order to compare with the cloud diagnostics from COSP, we also use the satellite data products generated especially for model evaluation from ISCCP (Pincus et al., 2012; Zhang et al., 2012), Moderate Resolution Imaging

Spectroradiometer (MODIS) (Pincus et al., 2012), and Cloud-Aerosol Lidar and Infrared Pathfinder Satellite Observation (CALIPSO) (Chepfer et al., 2010). Top-of-atmosphere cloud radiative effects are evaluated with the Clouds and the Earth's Radiant Energy System—Energy Balanced and Filled (CERES-EBAF v2.8) dataset (Loeb et al., 2012).

## 3 Results

In this section, we will focus on the results of June-July-August (JJA) and December-January-February (DJF), the two more

extreme seasons at the CONUS in a year when some long-standing systematic model errors are present.

## 3.1 Overall model performance

Taylor diagrams (Taylor, 2001) offer a concise way to summarize model performance and compare different model results. Here we employ Taylor diagrams to demonstrate the performance of a selection of important variables (Gleckler et al., 2016). Figure 2 shows the JJA and DJF model climatology of selected thermodynamic-related variables (numbered) over the

CONUS domain (i.e., the blue box in Fig. 1b) of the RRM grids. Green dots denote LRM results, whereas red dots HRM

results, and blue dots RRM results with the HRM parameters. The model results are illustrated relative to the verification data (marked by the reference point (1, 0)) described in Section 2.2. To make consistent comparisons between different model resolutions, model results are conservatively interpolated (with the "conserve" method of the Earth System Modeling Framework (ESMF, https://www.earthsystemcog.org/projects/esmf/) regridding software) to the coarser verification data grids before calculating the Taylor statistics. The radial axis shows the geographic variability (i.e., standard deviation (STD)) in the model climatology normalized by that in the observations. The angular axis indicates the spatial correlation (i.e., Pearson correlation coefficient (r)) between the simulations and the observations. By design, the distance to the reference point (1, 0) represents the centred root-mean-square (RMS) difference between the simulated and observed patterns normalized by the STD of the observations. The closer distance to the (1, 0) point, the better the model performance. We should note that the primary purpose of our analysis is to show how well the RRM, as an analogue to the HRM, reproduces the HRM results. The observations and the LRM results provide quantitative references to examine the RRM-HRM similarity and also identify poorly simulated behaviours as targets of HRM development.

Relative to the evaluation datasets (Fig. 2ab), the thermodynamic variables are generally well-simulated by EAMv1 with all three model configurations. All correlation coefficients are greater than 0.85 (mostly > 0.95). Normalized STDs lie close to the 1.0 dashed curve, especially in JJA. The model generally represents these large-scale circulation related quantities better with finer resolution settings, as the red dots (HRM) are usually closer to the (1, 0) point (representing evaluation data) than the corresponding green dots (LRM). Nevertheless, there are a few exceptions, for example, 2-meter air temperature ($T_{2m}$ or TREFHT) in JJA (see Fig. 2a), which is likely associated with cloud and thus surface radiation changes (Van Weverberg et al., 2018) along with feedbacks (surface energy partitioning shifting towards more sensible heat flux) from the land surface model.

More importantly, when using the HRM as the reference point (Fig. 2cd), blue dots (RRM) are located closer to (1, 0) than the green dots (LRM), indicating that the RRM mimics the HRM behaviours quite well. Additionally, we plot the RRM_LR results (purple dots) on Fig. 2cd to illustrate the potential impact of poor scale awareness, which is a common problem for current climate models, on conventional RRM applications. Lacking the tuned HRM parameters, previous RRM studies often heavily rely on LRM parameters and cannot quantify the likely performance deterioration due to the parameter-resolution mismatch. Here we take advantage of having both LR and HR tuned parameters to show the parameter influence on RRM performance. As expected, RRM_LR is generally less-satisfactory than RRM in matching the HRM behaviours (Fig. 2cd), but the extent varies for different quantities. For instance, the 200-hPa zonal wind (U200) is relatively insensitive to parameter changes in RRM configurations in both seasons. These results reflect the large-scale nature of upper troposphere wind fields. In contrast to U200, RRM total precipitable water (TMQ) shows greater sensitivity to parameter settings, since it is more closely related to sub-grid scale physical processes. These results suggest that RRM generally does well in representing large-scale thermo-dynamical behaviours of HRM, but some quantities are sensitive to the choice of

LRM or HRM parameter settings. A re-tuning may be needed for the refined region to optimize performance when model physical parameterizations are scale-sensitive. Specifically, for EAMv1, using the HRM parameter setting is recommended when one utilizes the RRM capability.

Compared to thermodynamics variables in Fig. 2, the cloud and precipitation variables in Fig. 3 are more sensitive to sub-grid scale parameterizations (e.g., convection, cloud microphysics, and radiation). Similar to Bacmeister et al. (2014), the variables in Fig. 3 are more poorly simulated than those in Fig. 2 in all configurations: they have weaker (0.5-0.9) correlation coefficients, and are further from the (1, 0) point in both seasons (Fig. 3ab). These results are consistent with the idea that improving the simulation of these variables requires both better resolved large-scale circulations and improved

representation of physical processes by better physical parameterizations. Figure 3cd demonstrates this idea quantitatively: the LRM-HRM differences become smaller for all variables after refining the CONUS grids (green to purple); the differences are further reduced by changing the parameters to match the HRM values (purple to blue). Nevertheless, our findings are similar: when increasing the resolution, model performance is generally better in DJF (Fig. 3b) and remains the same or slightly degraded in JJA (Fig. 3a); the RRM results follow those of the HRM closer than do the LRM in both

seasons (Fig. 3cd). In addition, all variables except total precipitation (PRECT) are more sensitive to the resolution change in winter than in summer (greater LRM-HRM separation in Fig. 3b than in Fig. 3a). For example, the variance of 500-hPa vertical velocity (OMEGA500) is almost independent of resolution in summer but about 50% larger in the HRM configuration than in the LRM in winter, suggesting stronger wintertime circulation and finer scale of resolved dynamics with the HRM configuration.

These overall Taylor statistics indicate that the RRM simulation with the HR parameters captures the HRM climatological statistics reasonably well, which provides the basis for potential applications of the RRM to effectively test physical parameterizations and simulate regional climate at high resolutions. In the following sections, we will further evaluate the similarity between the RRM and the HR EAMv1 simulations. We will examine some variables that are closely related to the

atmospheric hydrologic cycle with a primary focus on detailed aspects of precipitation, which remains a significant challenge in current climate models and is a major focus application for E3SM.

### 3.2 Regional geographic patterns

In this section we study whether RRM can reproduce the regional geographic patterns of hydrologic variables simulated by HRM.

### 3.2.1 Precipitation

Figure 4 shows the geographic pattern of mean total (large-scale + convective) precipitation differences between LRM and GPCP1DD observations, and the differences among model configurations over the CONUS domain in JJA. The differences

between the LRM and evaluation data (i.e., panel a) are computed on the evaluation data grid, while those between models (i.e., panels b-d) are computed on the HRM grid. Dotted regions mark where the differences are statistically significant at a 95% confidence level with the two-tailed Student's t-test assuming that each year is an independent sample. EAMv1 global precipitation results are described by Xie et al. (2018). Compared to GPCP1DD observations, the LRM mostly overestimates (up to 3 mm/day) western US precipitation, and underestimates (up to 4 mm/day) eastern and central US precipitation (Fig. 4a). As implied by the similar correlation coefficients of precipitation in Fig. 3a, the mean precipitation pattern exhibits rather uniform spatial changes (especially over land) among different model configurations (Fig. 4b-d). Over land, the HRM and RRM typically produce less precipitation than the LRM (partially due to the model tuning (not shown)) and the HRM rains the least. Differences between the RRM and the HRM are largely insignificant. In regions that pass the significance test, the differences are also relatively small, for instance, < 1 mm/day in the southern central US, and < 2 mm/day in the eastern US.

Figure 5 shows the differences in precipitation climatology patterns for DJF. An obvious change from the JJA results in Fig. 4 is the topographic signatures in differences between HRM and LRM (Fig. 5b), and RRM and LRM (Fig. 5c) over mountain regions in the western US, which are associated with better resolved topography in the RRM and HRM simulations. In addition, the signs of model differences (Fig. 5b-d) are less uniform in DJF than in JJA. Nevertheless, mean precipitation differences between the RRM and the HRM are also small (within ±2 mm/day), and not statistically significant over most grid cells. Overall, the RRM and HRM EAMv1 produce very similar mean precipitation geographic patterns in both seasons.

Following the COSP evaluation method described by Zhang et al. (2019), cloud fields (not shown here) from the LRM, HRM, and RRM are compared with the observations from ISCCP, MODIS, and CALIPSO. In JJA, all model configurations generally underestimate total cloud amount relative to CALIPSO observations over the CONUS. High thick (optical depth > 9.4) clouds lessen with enhanced horizontal resolution over the western central US, matching the precipitation change pattern over the same region in Fig. 4a. Low clouds along the western coast over the ocean increase noticeably in the RRM and HRM compared to the LRM. In DJF, greater reduction of cloud amount occurs at all levels at the western central US than in JJA with increased resolution. In both seasons, similar to precipitation, the RRM-HRM cloud differences are generally smaller than those for HRM-LRM.

### 3.2.2 Precipitable water

Figures 6 and 7 show the seasonal mean total precipitable water (TMQ) in JJA and DJF compared with the ERAI reanalysis data. The LRM underestimates the JJA TMQ over most places (see Fig. 6a) except the northwest US, and smaller areas of the eastern US and Mexico, where we observe significantly overestimated precipitation in Fig. 4a. As suggested by the improvement of precipitation with increasing resolution in Fig. 2, the LRM underestimation is generally improved in the

HRM (Fig. 6b) and the RRM (Fig. 6c). The mean RRM-HRM differences (Fig. 6d) are mostly positive ($< 4$ kg/m$^2$). This is due to reduction in precipitation in RRM than in HRM outside of CONUS, where the RRM resolution is coarser than that of the HRM.

5    In DJF, the LRM TMQ (Fig. 7a) resembles the patterns (overestimation over the western US and underestimation over the eastern US) of precipitation (Fig. 5a) against evaluation data. Such similarity implies that the precipitation biases in winter are directly related to flaws in precipitable water. The RRM and the HRM differ less (mostly statistically insignificant, see Fig. 7d) than their differences with the LRM (Fig. 7bc).

### 3.2.3 Low-level circulation

10    The low-level jet (LLJ) over the Great Plains of the US exerts significant impact on precipitation primarily in summer (Higgins et al., 1997; Pu and Dickinson, 2014). It is responsible for transporting about one-third of the moisture from the Gulf of Mexico to the central US (Helfand and Schubert, 1995). Based on reanalysis data, Higgins et al. (1997) reported connections between the Great Plains LLJ events and regional precipitation anomalies in summer, such as greater precipitation over the north central US and Great Plains and declined precipitation along the Gulf coast and east coast. Here, 15   we examine the 850-hPa horizontal wind speed (Figs. 8 and 9, the difference vectors are shown by colours (magnitudes) and magenta streamlines (directions)) as an example of the low-level circulation.

In summer, the LRM simulates stronger wind than the ERAI reanalysis over a large portion of CONUS, but weaker southerly LLJ at the central US (see Fig. 8a), which contributes to the low precipitation bias in the Great Plains and along the 20   Gulf coast in Fig. 4a. Enhancing resolution significantly strengthens the LLJ (Fig. 8bc), consistent with results presented by Berg et al. (2015) for reanalyses with a range of resolutions, and reduces the differences compared to the ERAI reanalysis, since simulations at finer horizontal resolution can resolve the LLJ-related temperature and pressure gradients better than ones at coarser resolution. Contrarily, the overestimation of zonal wind strength over the northeastern US becomes slightly worse with finer resolution. The RRM-HRM (Fig. 8d) difference (mostly within $\pm 0.8$ m/s) is generally smaller than that in 25   other panels, especially for the LLJ region over the south-central US.

In winter, we find about twice greater wind differences than in summer (note the different colour scales in Figs. 8 and 9). However, the main features remain unchanged, for instance, the LRM also simulates too strong (mostly $>1.0$ m/s) zonal winds (Fig. 9a), and the RRM-HRM difference is relatively small (within $\pm 2.0$ m/s) and mostly insignificant (Fig. 9d). These 30   results suggest that the RRM mimics the low-level circulation of the HRM, including the summertime LLJ. Together with the precipitable water results in the previous section, they imply similar water vapor transport patterns in the RRM and HRM. Therefore, the RRM is a useful tool to study the HR water transport over CONUS.

### 3.2.4 Surface air temperature

Warm and dry model biases over the summertime central US have been studied for more than a decade (Klein et al., 2006), and are still deficient in the current generation of regional and global climate models (Cheruy et al., 2014; Mueller and Seneviratne, 2014; Lin et al., 2017; Ma et al., 2018; Morcrette et al., 2018). Land (soil moisture)-atmosphere coupling plays a key role in causing warm and dry biases (Mo and Juang, 2003; Klein et al., 2006; Lin et al., 2017; Ma et al., 2018; Van Weverberg et al., 2018), and the related precipitation biases.

Figure 10 shows the mean JJA patterns of differences in $T_{2m}$ between the LRM and ERAI data and between three EAMv1 model pairs over CONUS. Over the central US, the LRM simulation exhibits statistically significant positive temperature (up to 3 K) biases throughout the area (see Fig. 10a), corresponding to precipitation low bias (Fig. 4a) in this region. As implied from the Taylor diagram (Fig. 2), performance degrades further (by up to about 4 K) with enhanced resolution. This warm bias can be roughly attributed to two separate sources (Ma et al., 2018): the evaporative fraction (EF) contribution and the radiation contribution which is primarily caused by excessive absorbed solar radiation at the surface. EF is defined as the fraction of the combined latent and sensible heat fluxes that are in latent form. Models with too low EF tend to use the radiative input to heat the surface instead of evaporating water. The larger bias in the HRM is because the EF contribution is a few times larger with enhanced resolution, while the radiation contribution remains almost unchanged. The noisy and large differences in Fig. 10bc over western and central mountain regions are likely associated with topographic differences at different resolutions. Figure 10d shows that the RRM-HRM differences are small (< 2 K) and statistically not significant, but robustly positive over the west coast and negative elsewhere.

Figure 11 shows the $T_{2m}$ results in DJF. The LRM (Fig. 11a) still suffers from warm bias over the central US, but it is less severe and much less widespread than in JJA. Over almost the entire eastern US, the LRM underestimates (by up to 4 K) $T_{2m}$. The HRM (Fig. 11b) and RRM (Fig. 11c) simulations appear better than the LRM over the Great Plains, the north central US, and the southeastern US. The RRM-HRM differences in Fig. 11d are again the smallest among all panels and statistically insignificant except for the southwestern US.

So far, we have demonstrated that the RRM capability reproduces the characteristics of hydrologic fields simulated in HRM. This proves that the RRM is a reliable testbed which can be used to effectively study and understand these model biases. Next, we will present further analysis on precipitation with RRM and compare it with HRM. Note that the hydrological cycle is a major focus of E3SM of which precipitation is the most important atmospheric variable.

### 3.3 Precipitation characteristics

### 3.3.1 Partitioning between large-scale vs. convective precipitation

Precipitation in climate models (e.g., EAMv1) consists of large-scale and convective components. Large-scale precipitation results from condensation due to resolved processes at the model grid resolution and is simulated by the microphysics scheme, while the convective precipitation results from unresolved sub-grid scale processes that are approximated by the deep convection parameterization. Poor partitioning between these two components manifests as errors in the vertical structure of latent heating which corrupts the dynamical response of the environment to convection. Accurately capturing the partitioning is challenging for climate models, which often overestimate the convective component (Lin et al., 2013; Yang et al., 2013). Thus, the partitioning between the large-scale and convective precipitation is an important evaluation metric for climate models. Although they can be clearly defined in the model, the two precipitation components are difficult to separate observationally in a manner comparable to the model. Thus, we only plot the model results for the ratio.

Figures 12 and 13 display the mean ratio of large-scale to total precipitation from EAMv1 models in JJA and DJF, respectively. As expected, convection is a more important source of precipitation in summer and at lower latitudes. The ratio of the large-scale precipitation increases with resolution in Figs. 12 and 13 because more precipitation can be resolved with finer resolution grids and thus classified as large-scale precipitation. Similar convective precipitation changes with resolution are reported by Bacmeister et al. (2014) for CAM4 and CAM5. Consequently, compared to the LRM, large-scale precipitation in the HRM and RRM is more prevalent (especially in the north) during the summer months (see Fig. 12bc) and is even more dominant during the winter months (see Fig. 13bc). In both seasons, the RRM matches the HRM overall distributions of the precipitation partitioning including some regional details, for example, the contour lines along the Sierra Nevada mountains in California in DJF.

### 3.3.2 Precipitation intensity distribution

Besides the mean precipitation pattern and partitioning between its large-scale and convective components, it is crucial to accurately represent the precipitation intensity distribution in a changing climate, because evidence suggests that extreme events, such as severe storms and flooding, will intensify due to the direct impact of global warming on precipitation (Trenberth, 2011; Seeley and Romps, 2014; Walsh et al., 2014). Like many other global climate models (e.g., Dai, 2006; Stephens et al., 2010; Pendergrass and Hartmann, 2014), Terai et al. (2017) showed that EAMv0 suffers from deficiencies in precipitation intensity over the globe, overestimating the frequency of light to moderate rain compared to the GPCP1DD data.

Figure 14 shows EAMv1 vs. GPCP1DD intensity functions over CONUS in JJA and DJF. Before aggregating the distribution, modelled precipitation rates are interpolated with the ESMF conservative regridding method to the same 1º x 1º

grids as GPCP1DD data. All datasets are averaged over daily intervals. The frequency is then counted in log-bins of precipitation rates on each grid. In this way, the frequency functions from datasets at different spatial and temporal resolutions become comparable. It is evident in Fig. 14 that EAMv1 still simulates excessive light precipitation (< 10 mm/day) with all three configurations in both JJA and DJF. As implied by the mean behaviours in Figs. 12 and 13, convective precipitation accounts for a larger fraction of the total in JJA than in DJF across the whole spectrum (not shown in Fig. 14). Total precipitation from the RRM (blue dots) is closer to the HRM (red dots) than to the LRM (green dots) in most bins. These results suggest that we can use the RRM as a testbed to address issues of intensity statistics over CONUS in the HR configurations of future EAM versions.

### 3.3.3 Diurnal cycle of summertime precipitation

Representing the correct timing and location of these convection episodes is of critical importance for precipitation prediction and hydrologic research (IPCC, 2012). Doing so requires the ability to capture many different meteorological phenomena. For example, summertime mesoscale convective complexes (MCCs) or systems (MCSs) contribute a significant amount of the total precipitation and play an important role in extreme precipitation events over the central US (Maddox, 1980; Carbone et al., 2002; Ashley et al., 2003; Tuttle and Davis, 2006), while disorganized convection strongly influences precipitation over the southeastern US (Dai et al., 1999; Bacmeister et al., 2014; Rickenbach et al., 2015). The diurnal cycle of precipitation is one important measure of a model's ability to reproduce these phenomena. For example, Bacmeister et al. (2014) used the diurnal cycle of precipitation to diagnose deficiencies in capturing the observed phase of MCSs over the central US in CAM5 with both LR and HR configurations.

Figure 15 illustrates the mean diurnal phase and magnitude patterns of maximum precipitation in JJA from the NEXRAD data and the EAMv1 simulations. The mean diurnal maximum is determined from the first harmonic of the Fourier series constructed from the hourly precipitation time series in each grid box. The phase (local time) of the maximum is indicated by colours, while the magnitude the saturation of the colour. The NEXRAD data (Fig. 15a) shows the distinct nocturnal (19:00—04:00 LT) peak over the central US. This nocturnal peak has been attributed to the eastward propagation of MCSs originating at the front range of Rocky Mountains in the afternoon (Riley et al., 1987; Dai et al., 1999; Carbone et al., 2002; Jiang et al., 2006; Dirmeyer et al., 2012). Unfortunately, no model configuration is successful at capturing this night time maximum. The RRM and HRM diurnal phases are similar and show modest improvement over the LRM in the sense that they have weaker amplitudes (lighter colours in panels c and d than in panel b) of incorrect diurnal cycles. The similarity between RRM and HRM indicates that RRM simulations will be valuable for understanding and addressing this important model bias.

To evaluate the known eastward propagation feature of the convection in this area, we average the JJA precipitation over four sub-regions: mountains, high plains, middle plains, and low plains, outlined by solid square boxes on Fig. 15a. Figure

16 shows the mean composite diurnal cycle in these sub-regions. We first calculate simple mean diurnal cycle from the hourly time series for each grid box. The first and second diurnal harmonics of the mean diurnal cycle — obtained using Fast Fourier Transform — are retained and adjusted to local time to generate the composite diurnal cycle. The composite lines plotted in Fig. 16 are averages of the composite diurnal cycle on each grid box within the sub-regions. In the NEXRAD measurements (Fig. 16a), there is a clear propagating pattern: the maximum emerges over the mountains (black) in the afternoon at 15:00, moves eastward and intensifies across the Great Plains, and reaches the middle (blue) and the low (green) plains in the night at 20:00 and 00:00, respectively. The three EAMv1 simulations (Fig. 16b-d) do not reproduce the convection propagation and miss the nocturnal precipitation peak. Although the HRM and the RRM show better skill than the LRM from the mountains to the high plains, these convective events are not strong enough (smaller magnitudes compared to observations) to sustain propagation further east.

The late afternoon rainfall peak over the southeast US is associated with disorganized convection (Bacmeister et al., 2014), a different mechanism than that over the central US. The red dashed lines in Fig. 16 denote the results for the southeast US. The diurnal cycle over the southeast US is generally well-simulated by the LRM, HRM, and RRM (panels b-d), but the time of peak precipitation is a few hours early, consistent with the experience of other models (Dai et al., 1999; Stratton and Stirling, 2012; Bechtold et al., 2014). More physically based improvements are needed to find a solution to the summertime diurnal cycle issue for precipitation over the CONUS, and the RRM provides an efficient testbed for parameterization testing. Previous studies (e.g., Bechtold et al., 2004; Stratton and Stirling, 2012; Bechtold et al., 2014) provide possible solutions for this issue of simulating the diurnal cycle of convective precipitation over land by modifying convective trigger procedures, entrainment, and convective closures. Our recent study (Xie et al., 2019) shows substantial improvement in the precipitation diurnal cycle in the LRM by employing a new convective trigger with a dynamic constraint on the convection onset, and with the capability of detecting moist instability above the boundary layer. We will apply the RRM testbed to extend the new convective trigger to the HRM and report the results in a future paper. This bias in the diurnal cycle of convection is significantly improved in convection-permitting (horizontal grid spacing < 2-4 km) simulations (Prein et al., 2015). The E3SM project is making progress in developing its convection-permitting version (E3SMv4), for which the RRM testbed will be heavily relied on.

## 4 Nudging capability for RRM

Nudging is an effective technique to create quasi-deterministic model realizations of observations for a specific time period. There is increasing use of nudging in climate model development and evaluation of physical parameterizations (e.g., Jeuken et al., 1996; Ghan et al., 2001; Kooperman et al., 2012; Zhang et al., 2014). Since nudging simulations constrain the model states closer to observed meteorological conditions, they facilitate evaluation of modelled physics during specific meteorological episodes. Therefore, nudging can help advance process-level understanding of physical phenomena, and

ultimately improve physical parameterizations. This is similar to the hindcast approach (Phillips et al., 2004; Ma et al., 2015) that has been widely used for climate model evaluation. EAMv1 has a built-in nudging capability as part of its physics module. When running the nudged RRM, one has various choices available such as nudging variables, locations, and time scales. In this section, we will provide an example of the value of EAMv1 RRM nudging simulations.

The EAMv1 nudging capability in the physics module allows relaxation of model state variables (U, V, T, and specific humidity, or a subset thereof) towards analysis/reanalysis data. The nudging strength is determined by a fractional nudging coefficient between zero and one, which can be a spatial constant or a spatial variable specified by a Heaviside window function. Following previous findings by Zhang et al. (2014) and Ma et al. (2015), we opt to only nudge horizontal velocities

for better cloud and aerosol properties with a 6-hour relaxation time scale (see Eq. 1 of Zhang et al. 2014). The nudging coefficient map is shown in Fig. 17. The corresponding nudging parameter settings are documented in Table 3. This non-US nudging setting creates a smooth transition from the strongest nudging (red) over coarser grid points to the weakest nudging (blue) over finer grid points. Running in this mode builds a pseudo-regional model framework in a global model. It gives the simulation more freedom over part of the HR region and reduces the nudging noise due to inconsistency between the model

and analysis data over the free-running region for better evaluation of physics over this region.

As an example of the nudging results, we create the Hovmöller diagrams (Fig. 18) of hourly mean total precipitation, meridionally averaged over 35ºN—45ºN, 93ºW—115ºW (the magenta box in Fig. 17) during the period of the DOE Atmospheric Radiation Measurement (ARM) Facility's Midlatitude Continental Convective Clouds Experiment (MC3E,

April 22 – June 6, 2011) (Jensen et al., 2016). The main science goal of the MC3E campaign is to improve the understanding of midlatitude continental convective cloud systems and their interactions with environment (Xie et al., 2014). Many cloud and precipitation events are observed and clearly shown from the NEXRAD panel (Fig. 18a), such as convective events on April 25 and around May 23, and widespread stratiform rain on May 10. As expected, the AMIP simulation (Fig. 18b) struggles to capture the statistics of these high-frequency weather systems. The RRM nudging simulation (Fig. 18c)

reproduces the timing and location of most events because nudging the horizontal velocities outside of the analysed area provides more realistic boundary conditions of the large-scale circulation in the free-running domain. There are still some deficiencies in the nudged simulation, for example the incorrect number and propagating speed of convective events, particularly after May 15. The nudged RRM has cleanly separated these remaining (model-deficiency based) problems from issues related to the large-scale circulation. This demonstrates that the nudged RRM is an effective testbed for isolating and

fixing parameterization problems at resolutions we cannot afford to run globally.

## 5 Summary and discussion

We have presented an overview of the climatological results comparing initial atmosphere-only simulations from globally uniform low-resolution (LR, 1º), high-resolution (HR, 0.25º), and regionally refined model (RRM, 1º to 0.25º) over the contiguous US (CONUS) with the atmosphere model version 1 (EAMv1) using the Energy Exascale Earth System Model (E3SM). Our analysis has established that the RRM can generally mimic HRM climate behaviour over the finely resolved portion (CONUS) for both well-simulated larger-scale thermodynamics fields and less-satisfactory smaller-scale physical variables.

Similar to other models (Dai, 2006; Bacmeister et al., 2014), the EAMv1 HRM suffers from deficiencies in convection, clouds, and moist physics (Xie et al., 2018). To verify that the RRM is a suitable alternative framework to the HRM to address these deficiencies, we examine the seasonal mean geographic patterns of precipitation, vertically-integrated precipitable water, low-level circulation, and surface temperature for JJA and DJF. Given its key importance in the atmospheric hydrologic cycle, we conduct in-depth analysis on precipitation, including fractions of the large-scale precipitation and daily intensity functions, and the JJA diurnal cycle. Overall, the RRM is similar to the HRM for many finer scale features, and including reproducing longstanding climate model biases, such as lack of summertime nocturnal precipitation peaks and the warm bias in surface air temperature.

Poor scale awareness of EAMv1 physical parameterizations necessitates retuning the model when increasing resolution. Different from previous RRM work using primarily the LR model (LRM) parameters, we make use of both LRM and HRM parameters and illustrate the significant impact of HR vs. LR parameters on RRM performance due to poor scale awareness, particularly for variables that are closely related to sub-grid scale physical processes. The high sensitivity of EAMv1 to model resolution suggests the need to develop better scale-aware physical parameterizations or convection-permitting simulations in the future. This study demonstrates how RRM can be used as a useful testbed to evaluate potentially improved schemes across different spatial scales.

To help users better utilize the E3SM RRM capability, we provide detailed guidance on running the RRM in the nudging mode so that deficiencies in model physical parameterizations can be better isolated. By relaxing the horizontal velocities over coarser resolution grids to analysis data, we create more realistic boundary conditions to the free-running higher resolution area. Such a pseudo-regional model framework within a global model displays great advantages in capturing observed convective episodes over the AMIP configuration, and hence allows us to calibrate simulated physical processes against observations under different meteorological conditions. With more realistic large-scale circulation conditions, the nudged RRM can be used as a physics testbed for regional process-level studies and aid in the development of future HR EAM versions.

*Code availability.* The E3SM source code is available on GitHub: https://github.com/E3SM-Project/E3SM.

*Author contributions.* QT and SAK designed the experiments. QT and WL performed the simulations and analyzed the data. QT, SAK, and
SX designed the scope and structure of the manuscript. QT prepared the manuscript with contributions from all co-authors.

*Acknowledgements.* This research was primarily supported as part of the Energy Exascale Earth System Model (E3SM) project and partially
supported by the Climate Model Development and Validation activity, Atmospheric System Research Program, and an earlier project
entitled Climate Science for a Sustainable Energy Future, funded by the U.S. Department of Energy (DOE), Office of Science, Office of
Biological and Environmental Research (BER) under the auspices of the U.S. DOE by Lawrence Livermore National Laboratory under
contract DE-AC52-07NA27344. The Pacific Northwest National Laboratory is operated for DOE by the Battelle Memorial Institute under
contract DE-A06-76RLO 1830. This paper has been authored by employees of Brookhaven Science Associates, LLC, under contract No.
DE-SC0012704 with the U.S. DOE. The publisher by accepting the paper for publication acknowledges that the U.S. Government retains a
non-exclusive, paid-up, irrevocable, world-wide license to publish or reproduce the published form of this paper, or allow others to do so,
for U.S. Government purposes. This research used resources of the National Energy Research Scientific Computing Center, a DOE Office
of Science User Facility supported by the Office of Science of the U.S. DOE under Contract No. DE-AC02-05CH11231. The E3SM
simulation data used in this study can be downloaded at http://portal.nersc.gov/project/acme/tang30/E3SMv1_RRM_CONUS/. The ERA-
Interim reanalysis data can be obtained from http://apps.ecmwf.int/. ISCCP cloud amount data are available from
https://isccp.giss.nasa.gov/products/browsed2.html. GPCP and GPCP1DD precipitation datasets can be obtained from
https://www.esrl.noaa.gov/psd/data/gridded/data.gpcp.html and ftp://meso.gsfc.nasa.gov/pub/1dd-v1.2/. NEXRAD data are available from
https://data.nodc.noaa.gov/cgi-bin/iso?id=gov.noaa.ncdc:C00345. CERES-EBAF top-of-atmosphere cloud radiative effects data can be
downloaded at https://ceres.larc.nasa.gov/products.php?product=EBAF-TOA. The satellite data for COSP are archived at
http://climserv.ipsl.polytechnique.fr/cfmip-obs/. The authors thank Christopher Terai for providing the GPCP1DD data. The authors also
thank J. J. Gourley and Zac Flamig of the National Severe Storms Laboratory for access to the archives of NEXRAD NMQ Q2/3 products
used in this study. LLNL-JRNL-764721.

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

(a)

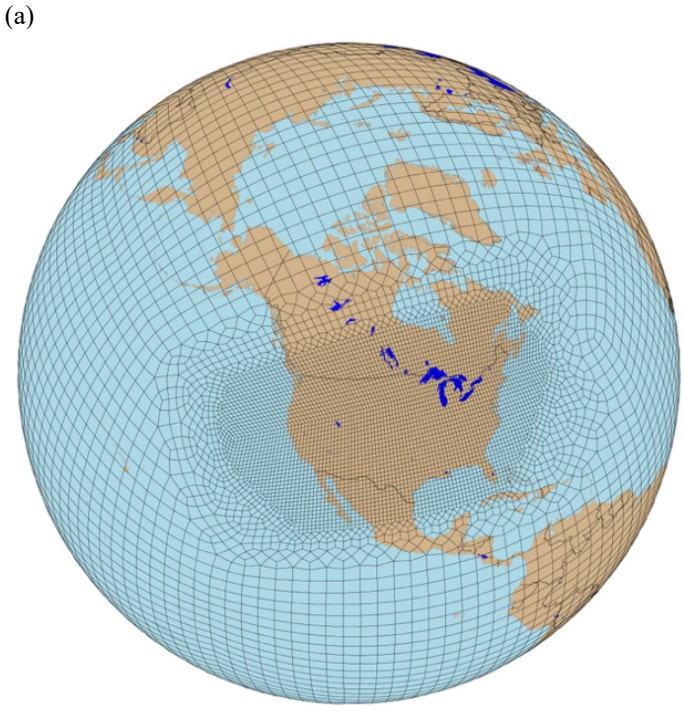

(b)

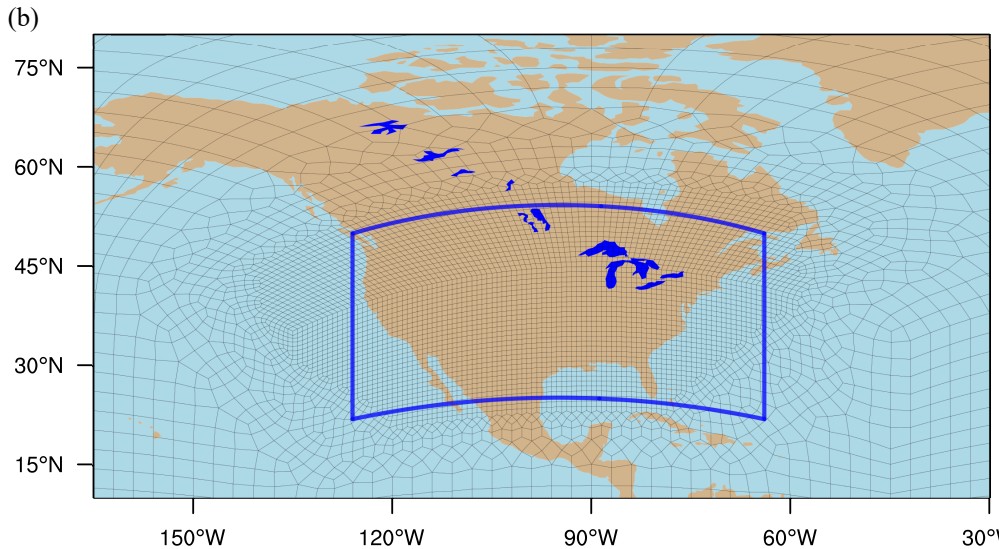

**Figure 1: The CONUS regionally refined grid in (a) a global orthographic projection and (b) a cylindrical equidistant projection zoomed in over the high-resolution (HR) portion. The effective resolutions for the low-resolution (LR) and the HR regions are 1°and 0.25°, respectively. The two regions are connected with a transient area. The blue box (latitude range: 22°N–50°N, longitude range: 64°W–126°W) in panel (b) represents the analysed area for CONUS.**

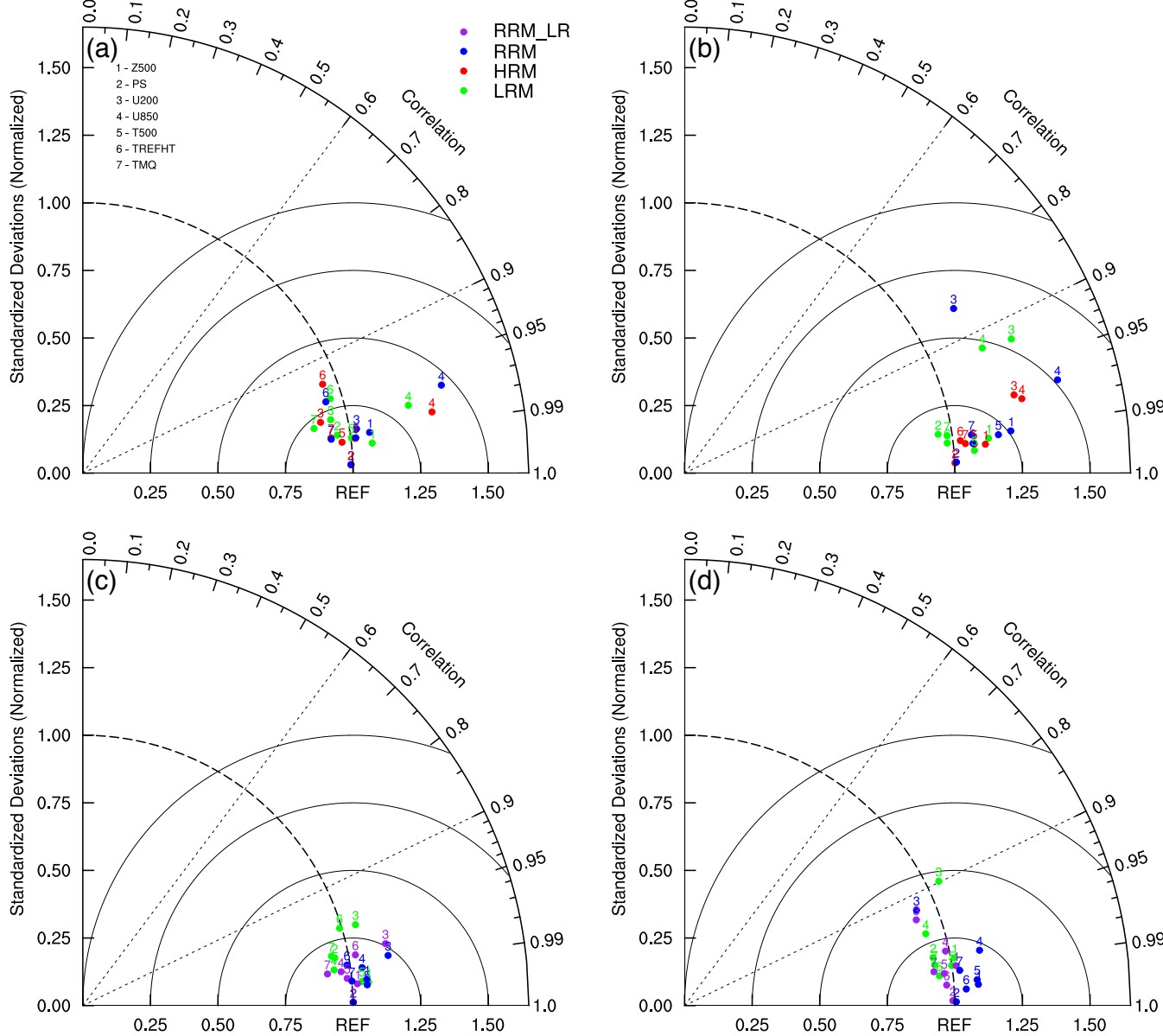

**Figure 2: Taylor diagrams for three different model climatologies (color-coded: green – LRM, red – HRM, blue – RRM, and purple – RRM_LR) in JJA (left column) and DJF (right column). Results are from the CONUS domain (the blue box in Fig. 1b). For panels (a) and (b), verification data are used as the reference point (1, 0), and statistics are calculated on the coarser verification grids. For panels (c) and (d), the HRM is the reference, and statistics are calculated on the HRM grids. The numbers represent: 1 – 500-hPa geopotential height (Z500), 2 – surface pressure (PS), 3 – 200-hPa zonal wind (U200), 4 – 850-hPa zonal wind (U850), 5 – 500-hPa temperature (T500), 6 – 2-meter air temperature (TREFHT or $T_{2m}$), and 7 – total precipitable water (TMQ).**

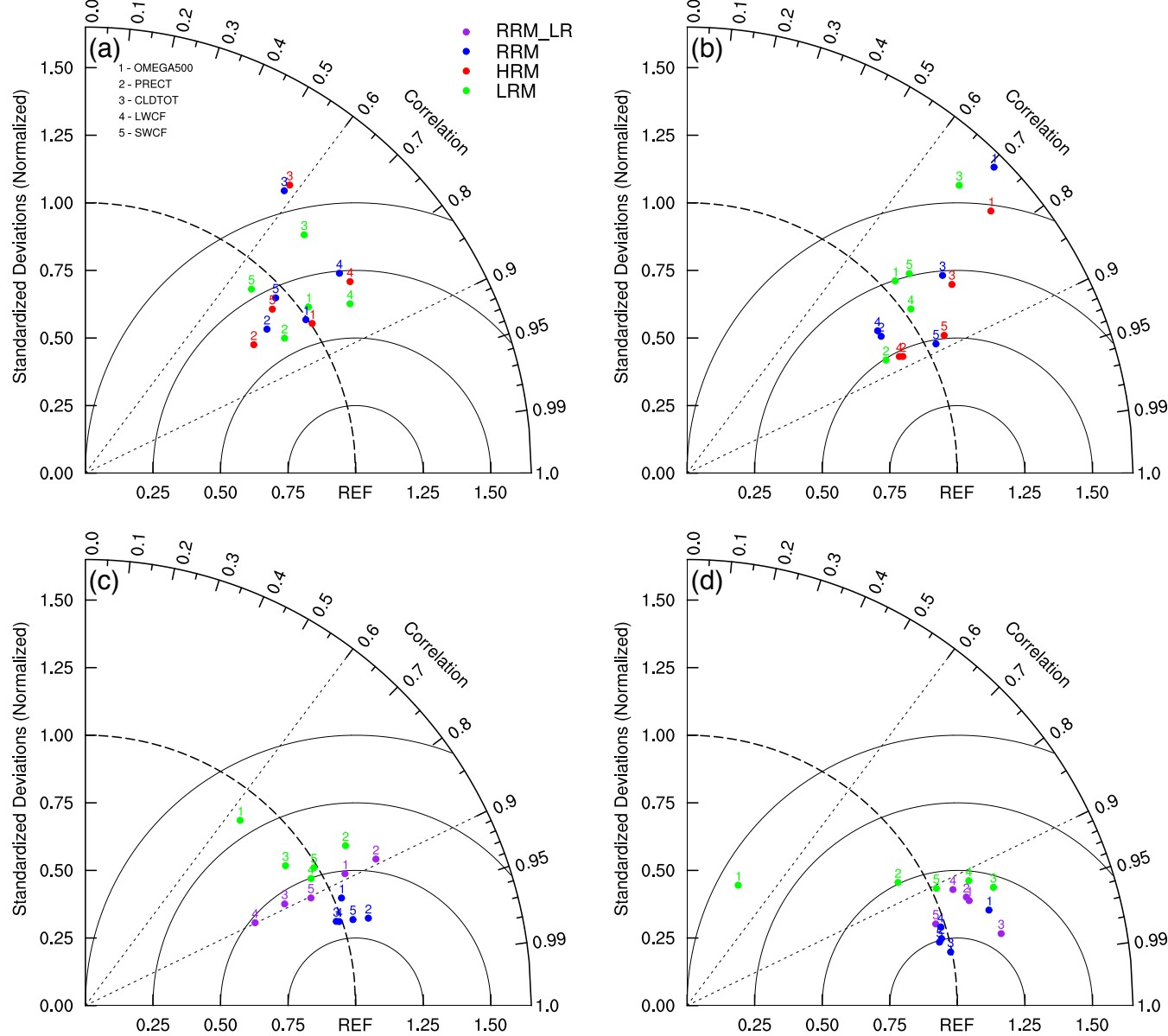

**Figure 3: Same as Fig. 2, but the numbers represent: 1 – 500-hPa vertical velocity (OMEGA500), 2 – total precipitation (PRECT), 3 – vertically-integrated total cloud fraction (CLDTOT), 4 – longwave cloud forcing (LWCF), and 5 – shortwave cloud forcing (SWCF).**

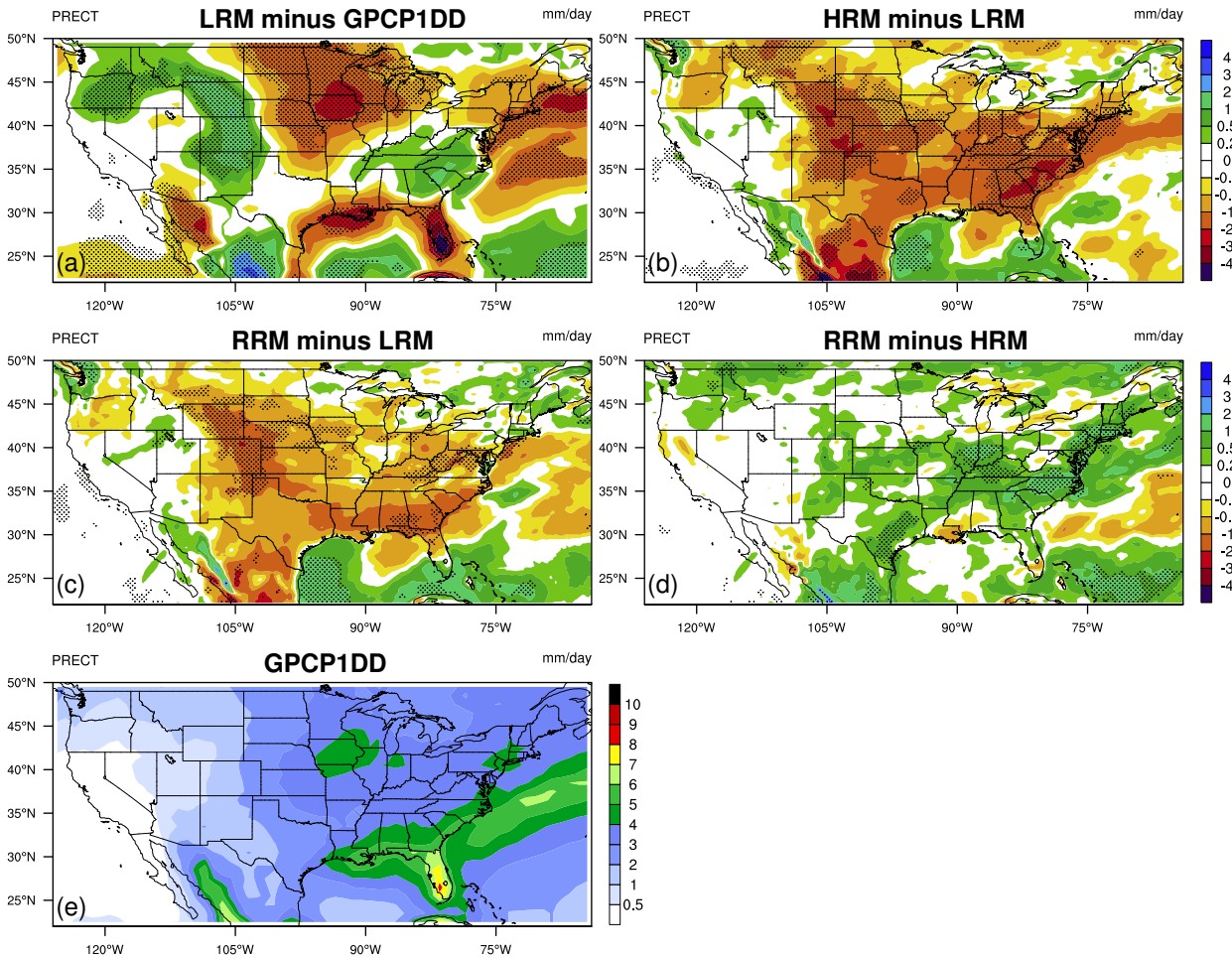

**Figure 4: Mean differences of total precipitation (unit: mm/day) in JJA for (a) LRM minus GPCP1DD data, (b) HRM minus LRM, (c) RRM minus LRM, (d) RRM minus HRM, and (e) GPCP1DD. The differences between the LRM and evaluation data (i.e., panel a) are computed on the evaluation grid, while those between models (i.e., panels b-d) on the HRM grid. Dotted areas denote where the differences are statistically significant at the 95% confidence level with the two-tailed Student's *t*-test.**

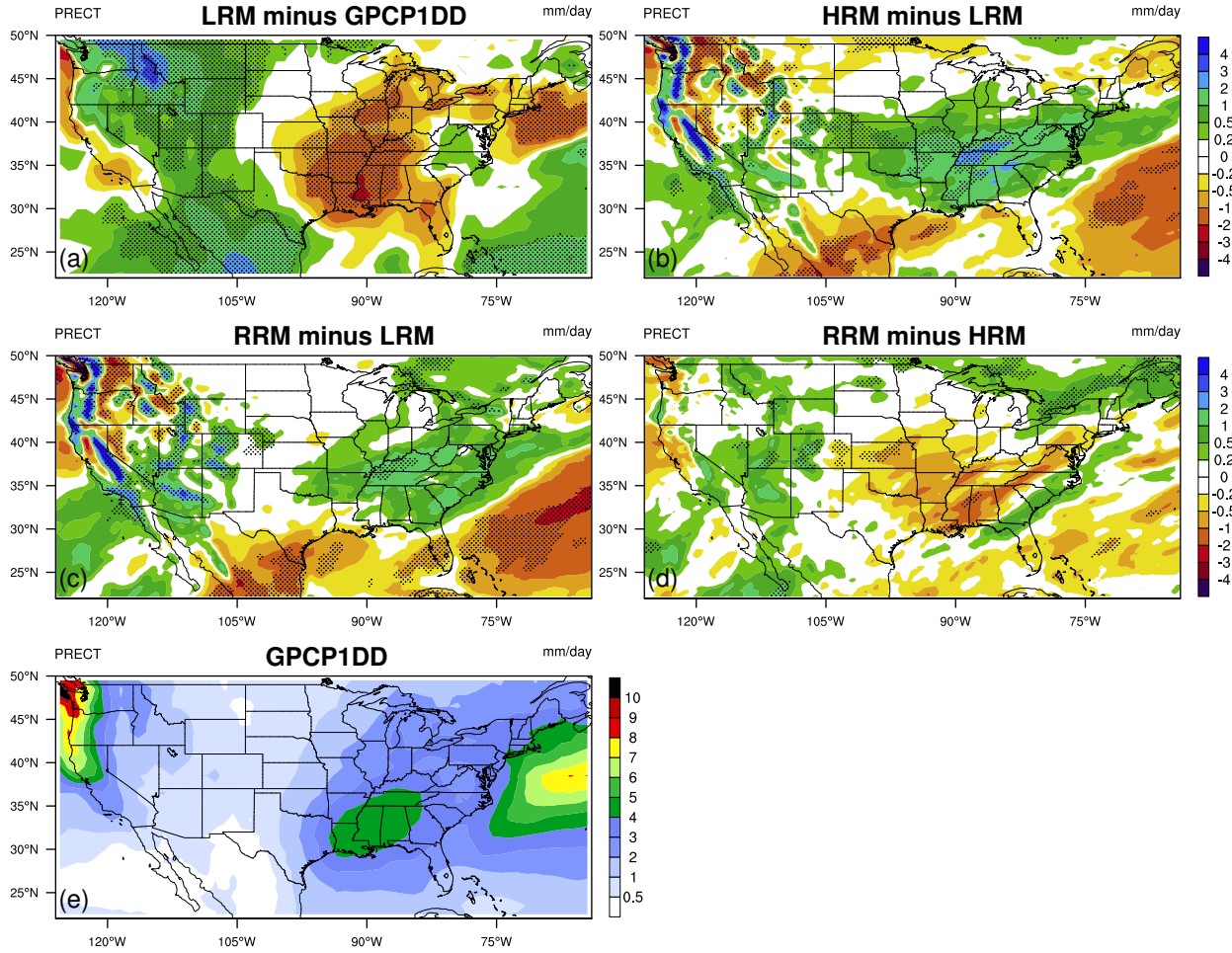

**Figure 5: Same as Fig. 4, but for DJF.**

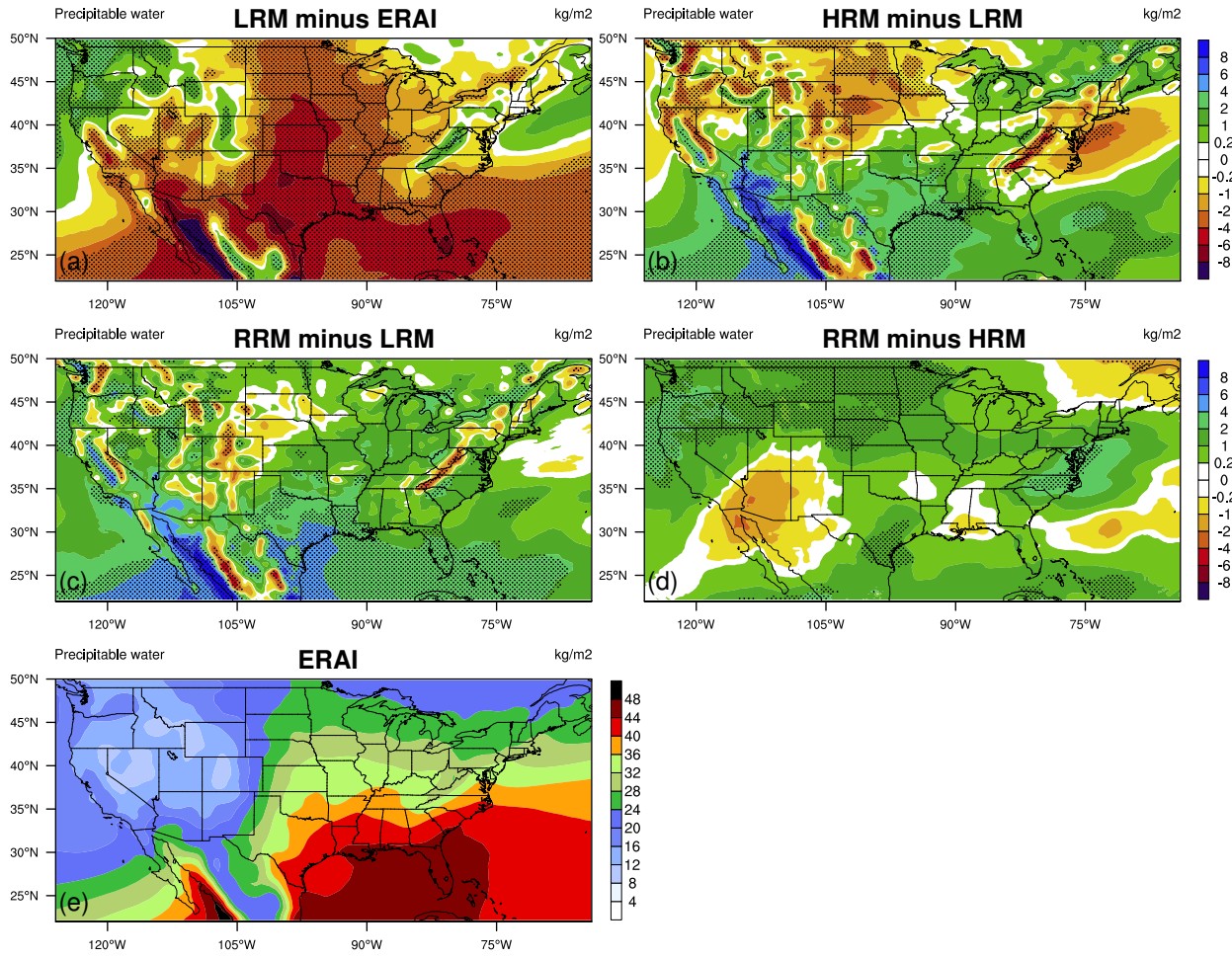

**Figure 6: Same as Fig. 4, but for total precipitable water (TMQ, unit: kg/m²) in JJA.**

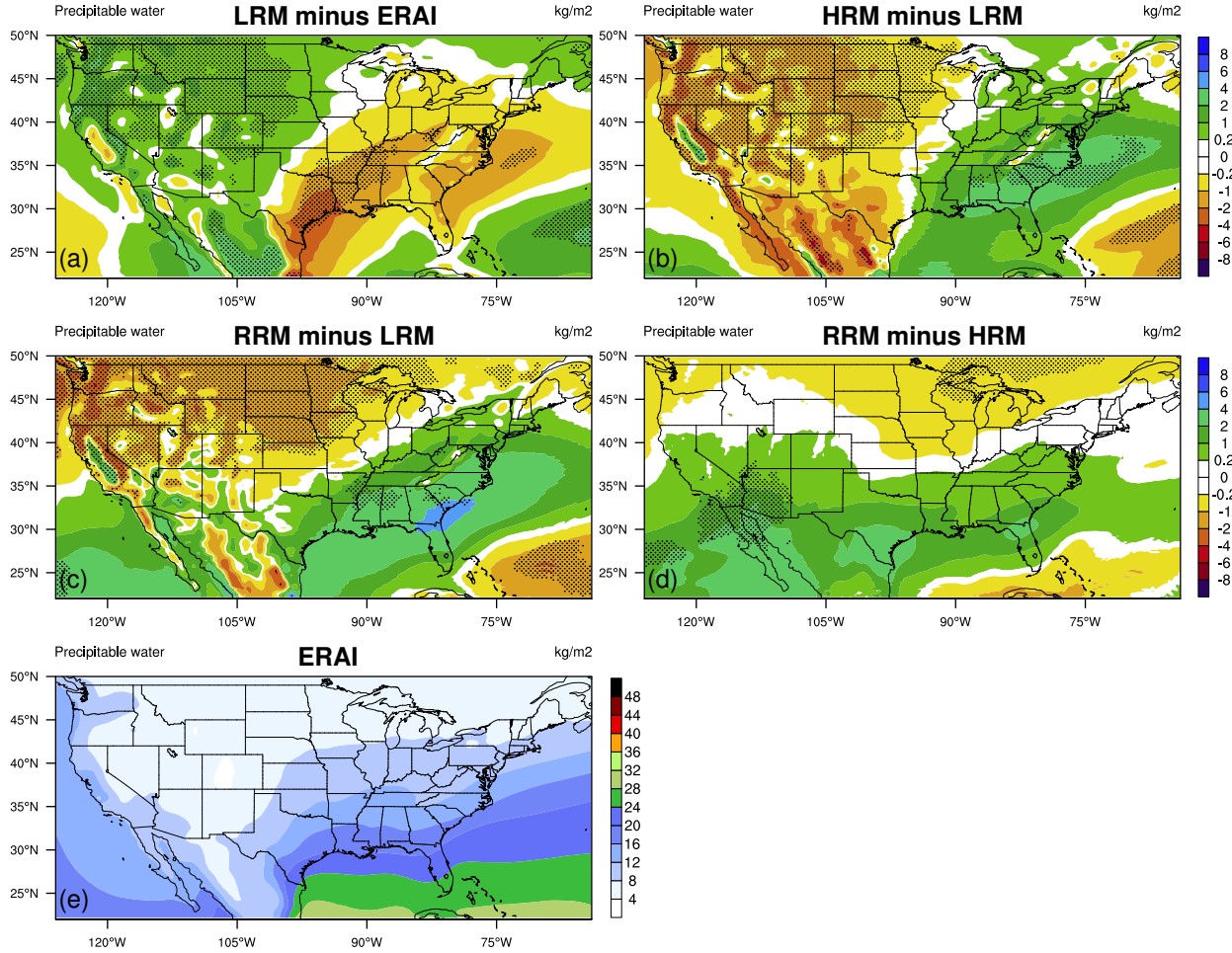

Figure 7: Same as Fig. 6, but for DJF.

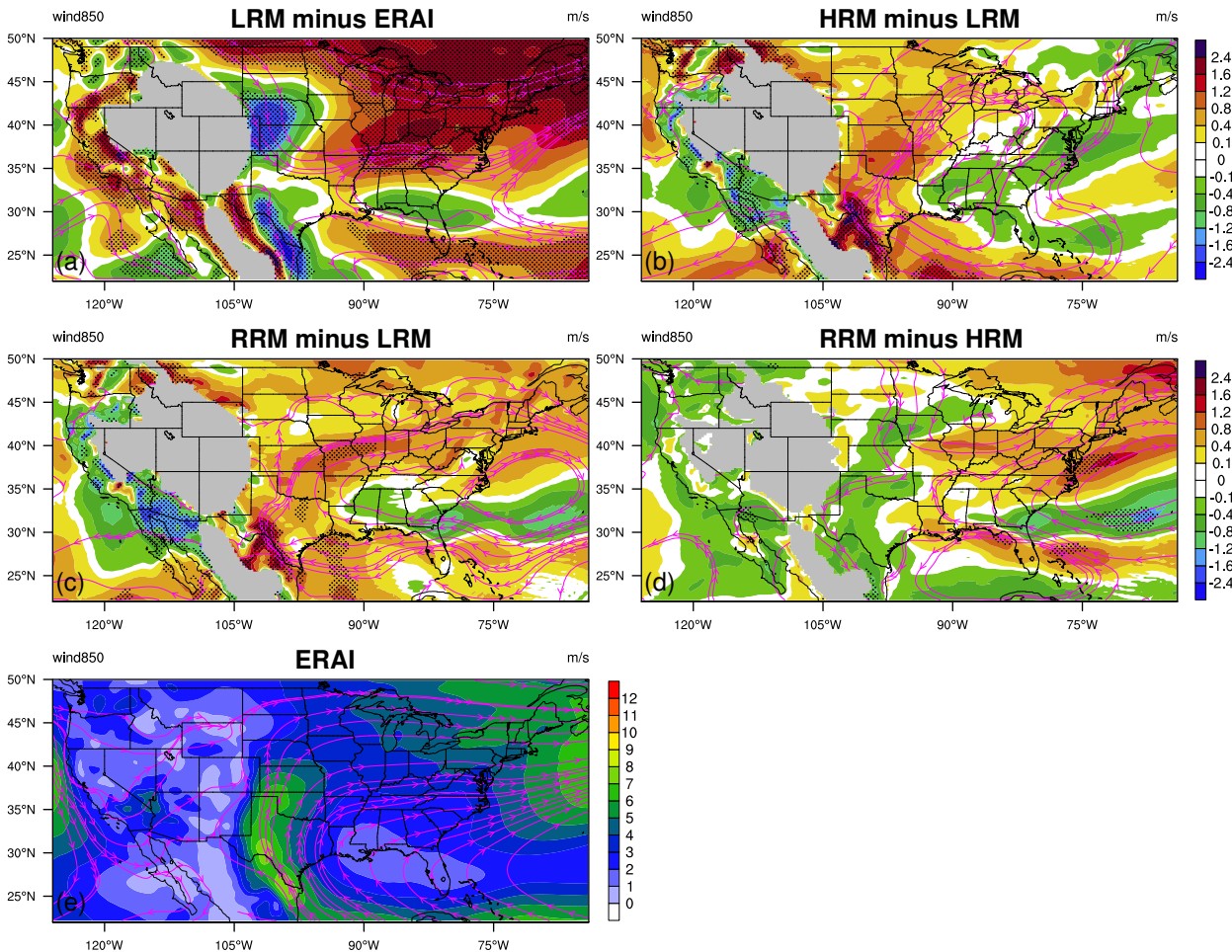

**Figure 8: Same as Fig. 4, but for 850-hPa wind speed (unit: m/s) in JJA. The vectors are shown by colours (magnitudes) and magenta streamlines (directions). Grid boxes where surface pressure is less than 850 hPa are shaded in gray on the difference plots.**

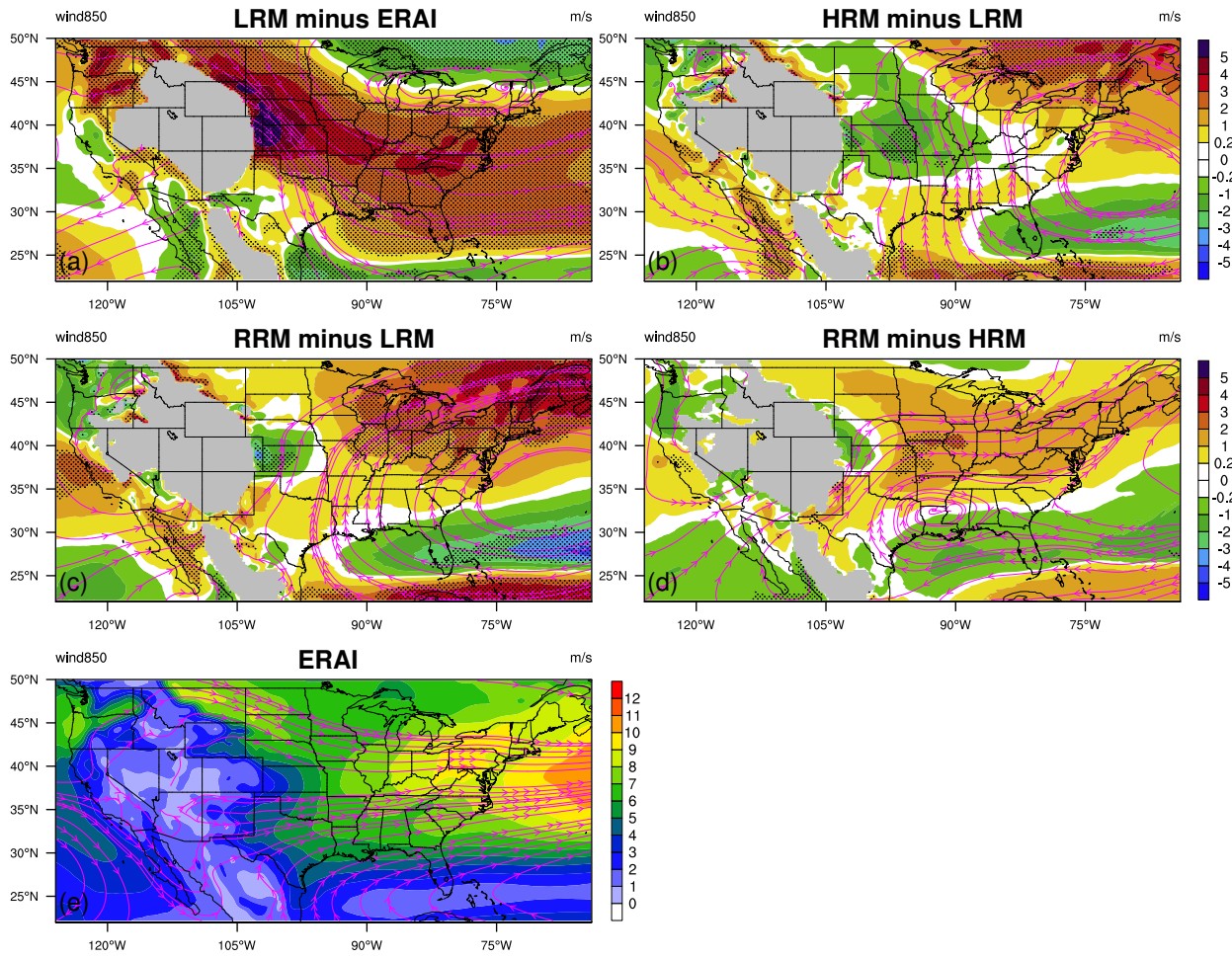

Figure 9: Same as Fig. 8, but for DJF. Note that the colour scale is different from that in Fig. 8.

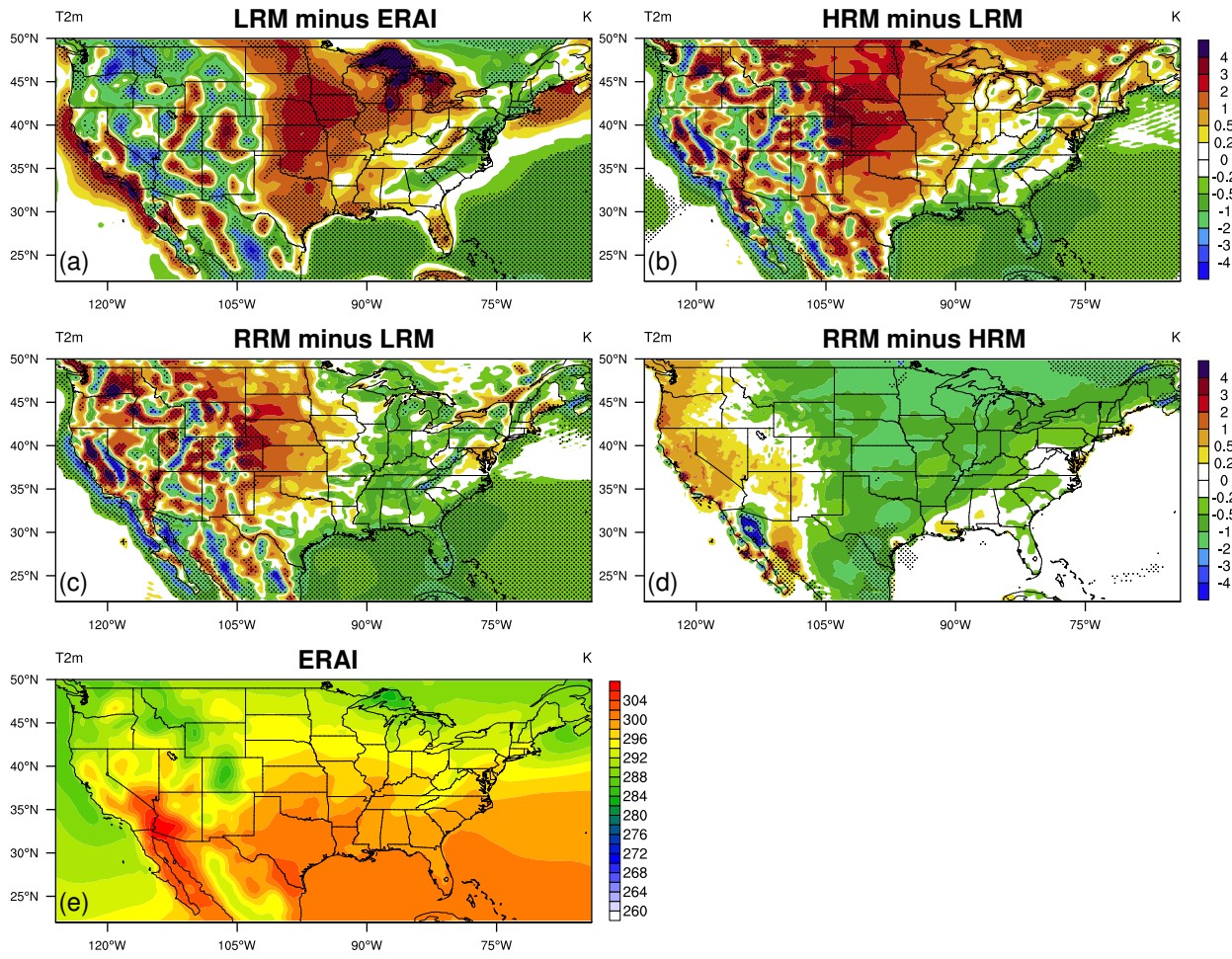

Figure 10: Same as Fig. 4, but for 2-meter air temperature (T$_{2m}$, unit: K) in JJA.

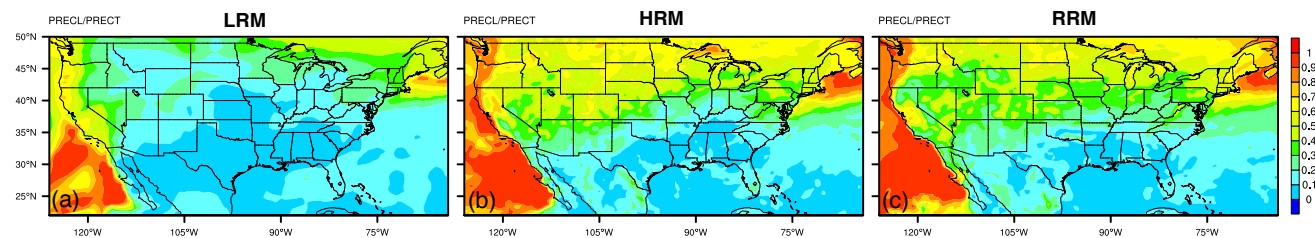

**Figure 11: Same as Fig. 10, but for DJF.**

**Figure 12: Mean ratio of large-scale to total precipitation in JJA for (a) LRM, (b) HRM, and (c) RRM.**

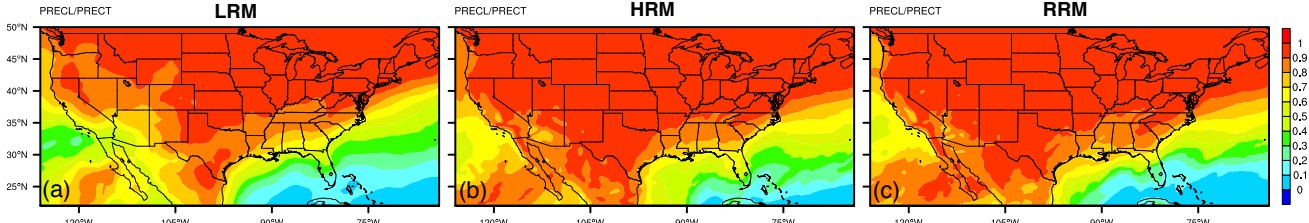

**Figure 13: Same as Fig. 12, but for DJF.**

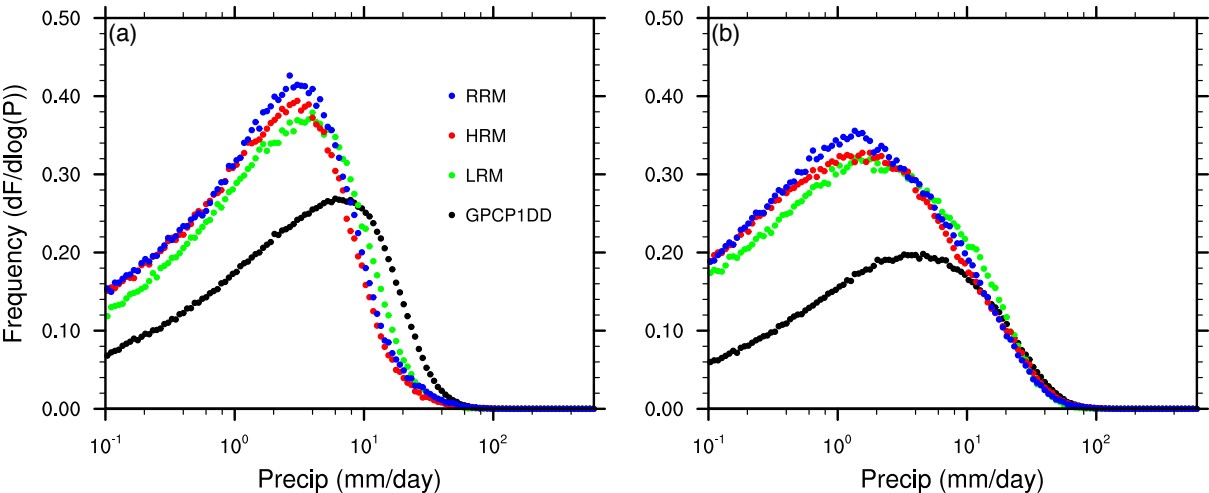

5    **Figure 14: Daily mean precipitation frequency (unit: dF/dlog(P)) functions of total precipitation for the GPCP1DD observation (black), and model simulations: LRM (green), HRM (red), and RRM (blue) in (a) JJA and (b) DJF. Before deriving the distribution, precipitation rates (unit: mm/day) are interpolated to 1º x 1º grids and averaged over daily intervals.**

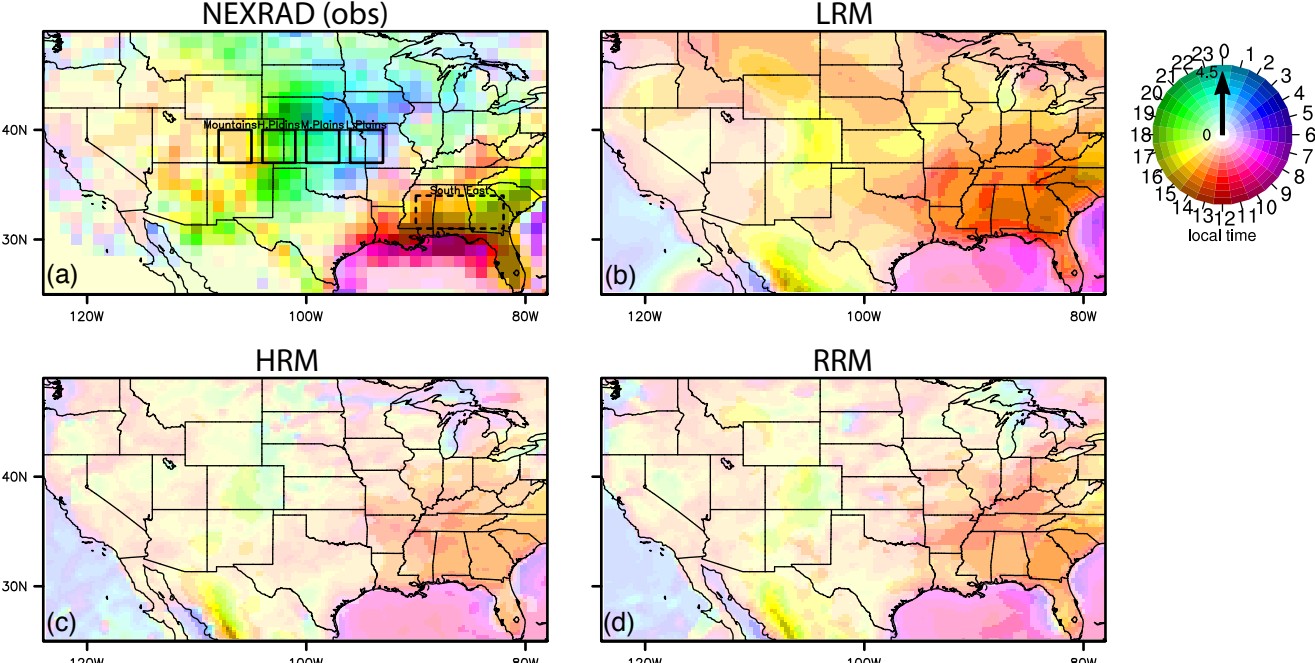

**Figure 15: Mean diurnal phase (local time, unit: hours) and magnitude (unit: mm/day) of the maximum precipitation in JJA calculated from the first harmonic for (a) NEXRAD observations, (b) LRM, (c) HRM, and (d) RRM. The phase is indicated by colours, while the magnitude the saturation of the colour. In panel (a), the solid boxes denote 4 central US regions from west to east: mountains (37ºN-40ºN, 105ºW-108ºW), high plains (37ºN-40ºN, 101ºW-104ºW), middle plains (37ºN-40ºN, 97ºW-100ºW), and low plains (37ºN-40ºN, 93ºW-96ºW). The dashed box marks the southeast (31ºN-34ºN, 82ºW-90ºW) regions.**

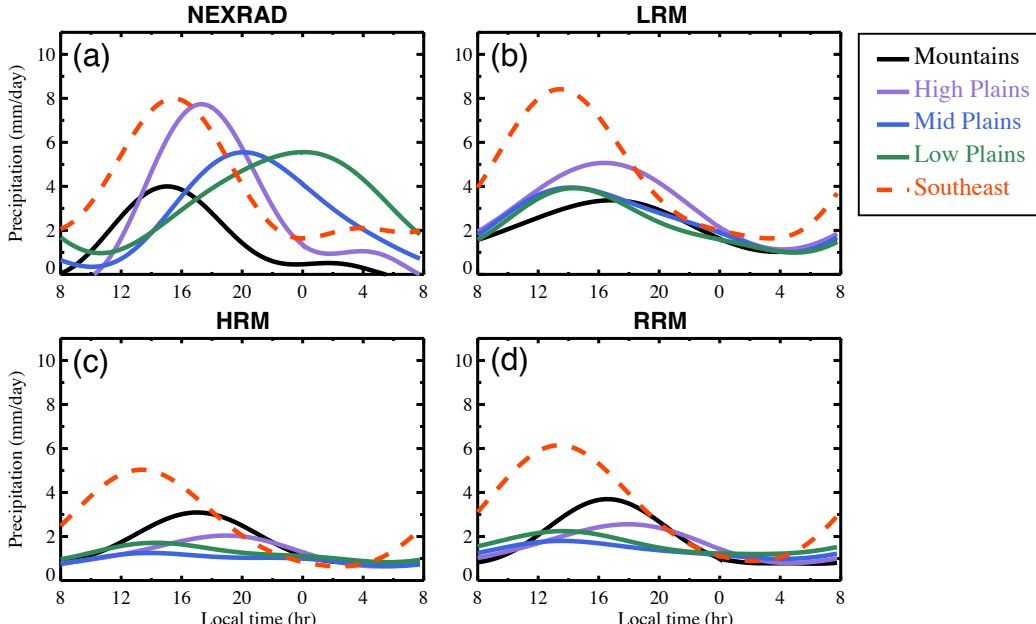

**Figure 16: Mean JJA composite precipitation diurnal cycle from the NEXRAD and model simulations for the sub-regions (denoted in Fig. 15): mountains (black lines), high plains (purple lines), middle plains (blue lines), low plains (green lines), and southeast (red dashed lines). Panels represent for (a) NEXRAD, (b) LRM, (c) HRM, and (d) RRM. We first calculate simple mean diurnal cycle from the hourly time series for each grid box. The first and second diurnal harmonics of the mean diurnal cycle — obtained using Fast Fourier Transform — are retained and adjusted to local time to generate the composite diurnal cycle. The composite lines plotted here are averages of the composite diurnal cycle on each grid box within the sub-regions.**

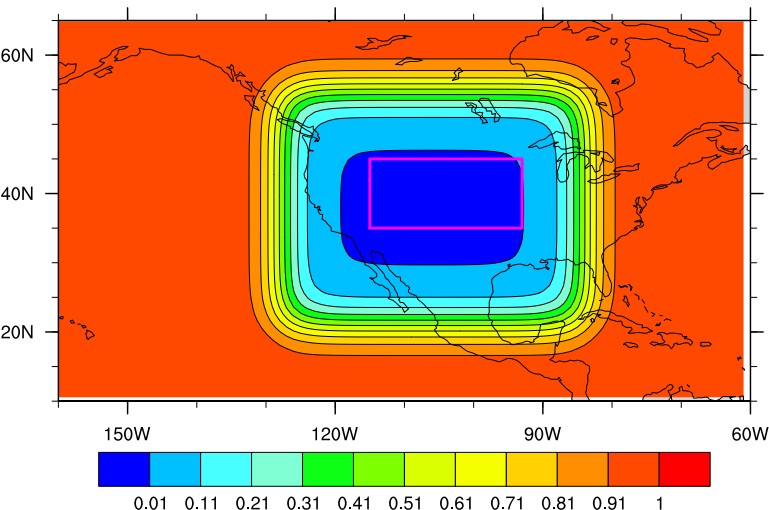

**Figure 17: Nudging coefficient map zoom-in over North America. A coefficient of 0 represents that no nudging is applied. The magenta box marks the area of the Hovmöller plots in Fig. 18.**

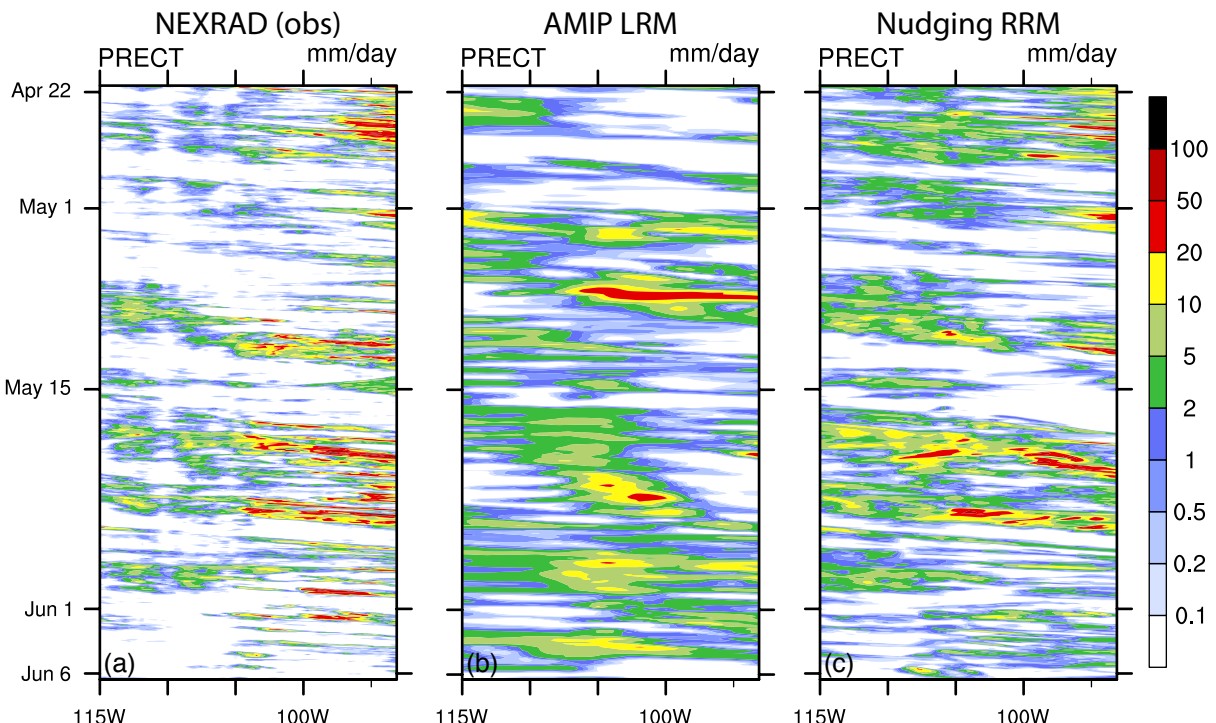

**Figure 18: Hovmöller plots of hourly mean total precipitation (unit: mm/day) over 35ºN—45ºN, 93ºW—115ºW during April 22 – June 6, 2011 for (a) NEXRAD observations, (b) AMIP LRM, and (c) nudging RRM.**

**Table 1: List of EAMv1 simulation configurations, speed, and costs. The speed and cost are for the NERSC Cori-KNL machine.**

| Simulation | Configuration | Effective angular resolution | Number of elements | Speed (SYPD) | Number of nodes | Cost (core-hours/year) |
|---|---|---|---|---|---|---|
| Low-resolution model (LRM) | Default | 1° | 5400 | 6 | 81 | 22,000 |
| High-resolution model (HRM) | Default | 0.25° | 86400 | 2 | 675 | 551,000 |
| Regionally refined model (RRM) | HRM default | 1° to 0.25° | 9905 | 1.7 | 88 | 84,000 |
| RRM_LR | LRM default | 1° to 0.25° | 9905 | 1.9 | 88 | 75,000 |
| LRM AMIP | Default | 1° | 5400 | 6 | 81 | 22,000 |
| RRM non-US nudging | HRM default | 1° to 0.25° | 9905 | 1.7 | 88 | 84,000 |

**Table 2: Summary list of observational and reanalysis-based evaluation datasets for model performance.**

| Variable | Data source | Period | Reference | Web link |
|---|---|---|---|---|
| Z500, PS, U200, U850, T500 $T_{2m}$, TMQ, RH500, OMEGA500 | ERA-Interim reanalysis | 1989—2005 | (Dee et al., 2011) | http://apps.ecmwf.int |
| CLDTOT | ISCCP | 1983—2001 | (Rossow and Schiffer, 1991) | https://isccp.giss.nasa.gov/products/browsed2.html |
| PRECT | GPCP | 1979—2009 | (Huffman et al., 2009) | https://www.esrl.noaa.gov/psd/data/gridded/data.gpcp.html |
| PRECT | GPCP1DD | 1997—2013 | (Huffman et al., 2001) | ftp://meso.gsfc.nasa.gov/pub/1dd-v1.2 |
| PRECT | NEXRAD | 2009—2013 | (NOAA, 2013; Giangrande et al., 2014) | https://data.nodc.noaa.gov/cgi-bin/iso?id=gov.noaa.ncdc:C00345 |
| LWCF, SWCF | CERES-EBAF | 2000—2013 | (Loeb et al., 2012) | https://ceres.larc.nasa.gov/products.php?product=EBAF-TOA |
| FISCCP1_COSP | ISCCP | 1983—2008 | (Pincus et al., 2012) | http://climserv.ipsl.polytechnique.fr/cfmip-obs |
| CLMODIS | MODIS | 2002—2010 | (Pincus et al., 2012; Zhang et al., 2012) | http://climserv.ipsl.polytechnique.fr/cfmip-obs |
| CLDTOT_CAL, CLDHGH_CAL, CLDMED_CAL, CLDLOW_CAL | CALIPSO | 2006—2010 | (Chepfer et al., 2010) | http://climserv.ipsl.polytechnique.fr/cfmip-obs |

**Table 3: Nudging parameter settings for the non-US nudging simulation.**

| Nudging parameter | Value |
| --- | --- |
| Nudge_Model | .true. |
| Nudge_Path | Path to analysis/reanalysis data |
| Nudge_File_Template | 'interim_se_%y%m%d00_%y%m%d18_TQUV-%s.nc' |
| Nudge_Times_Per_Day | 4 |
| Model_Times_Per_Day | 96 |
| Nudge_Uprof | 2 |
| Nudge_Ucoef | 1.00 |
| Nudge_Vprof | 2 |
| Nudge_Vcoef | 1.00 |
| Nudge_Tprof | 0 |
| Nudge_Tcoef | 0.00 |
| Nudge_Qprof | 0 |
| Nudge_Qcoef | 0.00 |
| Nudge_PSprof | 0 |
| Nudge_PScoef | 0.00 |
| Nudge_Beg_Year | 2011 |
| Nudge_Beg_Month | 1 |
| Nudge_Beg_Day | 1 |
| Nudge_End_Year | 2011 |
| Nudge_End_Month | 12 |
| Nudge_End_Day | 31 |
| Nudge_Hwin_lo | 1.0 |
| Nudge_Hwin_hi | 0.0 |
| Nudge_Hwin_lat0 | 38.0 |
| Nudge_Hwin_latWidth | 34.0 |
| Nudge_Hwin_latDelta | 3.8 |
| Nudge_Hwin_lon0 | 254.0 |
| Nudge_Hwin_lonWidth | 44.0 |
| Nudge_Hwin_lonDelta | 3.8 |
| Nudge_Vwin_lo | 0.0 |
| Nudge_Vwin_hi | 1.0 |
| Nudge_Vwin_Hindex | 73.0 |
| Nudge_Vwin_Hdelta | 0.1 |
| Nudge_Vwin_Lindex | 0.0 |
| Nudge_Vwin_Ldelta | 0.1 |

**Table 4: Acronym list.**

| | |
|---|---|
| ACME | Accelerated Climate Modeling for Energy |
| AMIP | Atmospheric Model Intercomparison Project |
| ARM | Atmospheric Radiation Measurement |
| CALIPSO | Cloud-Aerosol Lidar and Infrared Pathfinder Satellite Observation |
| CAM | Community Atmosphere Model |
| CERES-EBAF | Clouds and the Earth's Radiant Energy System—Energy Balanced and Filled |
| CESM | Community Earth System Model |
| CLUBB | Cloud Layers Unified By Binormals |
| CONUS | Contiguous United States |
| COSP | Cloud Feedback Model Intercomparison Project Observation Simulator Package |
| CSSEF | Climate Science for a Sustainable Energy Future |
| DJF | December-January-February |
| DOE | Department of Energy |
| E3SM | Energy Exascale Earth System Model |
| EAM | E3SM Atmosphere Model |
| EF | Evaporative Fraction |
| ENA | Eastern North Atlantic |
| ERAI | European Centre for Medium-range Weather Forecasting Interim |
| ESMF | Earth System Modeling Framework |
| GPCP | Global Precipitation Climatology Project |
| GPCP1DD | GPCP one-degree daily |
| HR | High-resolution |
| HRM | High-resolution Model |
| ISCCP | International Satellite Cloud Climatology Project |
| JJA | June-July-August |
| KNL | Knights Landing |
| Linoz | Linearized ozone chemistry |
| LLJ | Low-Level Jet |
| LR | Low-resolution |
| LRM | Low-resolution Model |
| MAM | Modal Aerosol Module |
| MCC | Mesoscale Convective Complex |
| MC3E | Midlatitude Continental Convective Clouds Experiment |
| MCS | Mesoscale Convective System |
| MODIS | Moderate Resolution Imaging Spectroradiometer |
| NERSC | National Energy Research Scientific Computing Center |
| NEXRAD | Next-Generation Radar |
| OMEGA500 | 500-hPa Vertical Velocity |
| PRECT | Total Precipitation |
| RMS | Root-Mean-Square |
| RRM | Regionally Refined Model |
| STD | Standard Deviation |
| SST | Sea Surface Temperature |
| TMQ | Total Precipitable Water |
| TREFHT | Reference Height Temperature |

| | |
|---|---|
| TWP | Tropical Western Pacific |
| U200 | 200-hPa Zonal Wind |
| US | United States |
| VR | Variable-resolution |

**Appendix A**

**Table A1: EAMv1 simulation setup details. *Used non-default parameter values in Table A2.**

| Simulation | Code hash | Grid | Compset |
|---|---|---|---|
| Low-resolution model (LRM) | 7a17edbe5 | ne30_ne30 | FC5AV1C-04P2 |
| High-resolution model (HRM) | 66793a1d3 | ne120_ne120 | FC5AV1C-H01A |
| Regionally refined model (RRM) | 7a17edbe5 | conusx4v1_conusx4v1 | FC5AV1C-04P2* |
| RRM_LR | 7a17edbe5 | conusx4v1_conusx4v1 | FC5AV1C-04P2 |
| LRM AMIP | dd18fc56e | ne30_oECv3 | F20TRC5-CMIP6 |
| RRM non-US nudging | 7a17edbe5 | conusx4v1_conusx4v1 | FC5AV1C-04P2* |

**Table A2: Non-default parameter values.**

| Parameter | Value |
|---|---|
| cldfrc_dp1 | 0.03 |
| clubb_c14 | 1.75 |
| clubb_c8 | 4.73 |
| rsplit | 2 |
| se_nsplit | 6 |
| cld_macmic_num_steps | 3 |
| zmconv_alfa | 0.2 |
| zmconv_c0_lnd | 0.0035 |
| zmconv_c0_ocn | 0.0043 |
| zmconv_dmpdz | -0.2e-3 |
| zmconv_ke | 5.0e-6 |

