# Peer review of "Regionally refined testbed in E3SM Atmosphere Model Version 1 (EAMv1) and applications for high-resolution modelling"

_Geoscientific Model Development, 2019_

## Referee Comment (RC1) · Anonymous Referee #1 · 15 Apr 2019

**# Comments on the manuscript**

Regionally refined capability in E3SM Atmosphere Model Version 1 (EAMv1) and applications for high-resolution modelling

Submitted by: Qi Tang, Stephen A. Klein, Shaocheng Xie, Wuyin Lin, Jean-Christophe Golaz, Erika L. Roesler, Mark A. Taylor, Philip J. Rasch, David C. Bader, Larry K. Berg, Peter Caldwell, Scott Giangrande, Richard Neale, Yun Qian, Laura D. Riihimaki, Charles S. Zender, Yuying Zhang, and Xue Zheng

**General Comments**

In the present manuscript the authors analyse whether a globally refined model (RRM) can be used instead of a globally high-resolution model (HRM). The RRM is computationally less demanding than the HRM. The authors compare different atmospheric quantities of simulations with the RRM, the HRM and a coarse grid global model (LRM). The RRM was run in to different configurations. One using the LRM physics and the other the HRM physics. The authors conclude that the RRM is a useful tool for high-resolution developments.

The manuscript needs major additions and major revisons (see specific comments below). After taking care of these the manuscript may be published in GMD.

**Specific Comments**

**Comments regarding the GMD principal criteria**

**Scientific significance - Excellent**

I am not aware of any publication that analyses in detail the effect of coarse grid global model, fine grid global model, coarse grid model with regional refined grid and tests of different physics tuning in the regional refinement.

**### Scientific quality - Good**

The overall scientific quality is good. A major deficiency is that the cited literature in the manuscript is heavily "American" weighted. The authors should increase their literature study of articles outside of the U.S.. They would have been aware of the remarks under item 4 in "Other specific comments" below.

**Scientific reproducibility - Poor**

The link to the source code of the model is given at the end of the manuscript https://github.com/E3SM-Project/E3SM. However, this information is not enough to reproduce the science by a fellow scientist. What at least is missing:

- exact information where to get the observational and reanalysis data from. This can

be provided by web links in table 2. Even this may be not enough, therefore comments on what has to be taken into account additionally should be added if needed.

- detailed technical model simulation setup (can be put into the Appendix). Actually, every information that is needed to run the model simulations as the authors did.

- post processing tools used for the statistical analyses

- time period(s) of simulations (see item 3 under "Other specific comments" below.

**presentation quality - Good**

The presentation of the results is overall good in a well-structured way. However, it misses some important information (see "Other specific comments" below)

**Other specific comments**

1. The authors claim to use a "fully-coupled Earth system model", however, in "2.1 Model overview and experiment design" they describe only the atmpospheric part. What about the ocean? What about the coupling?

2. page 5, line 7-8 "The LRM and HRM physics time steps are 30 minutes and 15 minutes, respectively". What about the dynamic time step? The dynamic time step depends on the grid width. How do the authors set the time step in the coarse grid region of the RRM and that one in the high-resolution domain?

3. Page 5, line 30 -34 "All free- running simulations are run for a period of 5 years. The first year is considered spin-up, thus we study the results from the last 4 years. The nudging run simulates year 2011, whereas the AMIP results are extracted for year 2011 from a long simulation starting from 1870 (Golaz et al., 2019). Model output is stored as monthly and hourly averages." This is the only part in the manuscript where I can find an information on the time period. From Table 2 it seems that all observations have different time periods. Are the time periods listed there are also the time periods which are compared with the model results?

4. Page 13, line 9-11 "... incorrect diurnal cycles. The similarity between RRM and HRM indicates that RRM simulations will be valuable for understanding and addressing this important model bias." and line 28-31 "... the time of peak precipitation is a few hours early, consistent with the experience of other models. More physically based improvements are needed to find a solution to the summertime diurnal cycle issue for precipitation over the CONUS, and the RRM provides an efficient tool for parameterization testing."

As the authors state right this is a well-known problem of coarse grid models. However, they did not cite any papers on possible solutions.

A possible solution for the ECMWF convection scheme is described in BECHTOLD, P., SEMANE, N., LOPEZ, P., CHABOUREAU, J.-P., BELJAARS, A., BORMANN, N., 2014: Representing Equilibrium and Nonequilibrium Convection in Large-Scale Models. Journal of the Atmospheric Sciences 71, (Heft 2), S. 734–753.

In general this problem does not appear in convection permitting simulations ( < about 2-4 km grid width). A starting point for reading on the effect of convective permitting simulations on the diurnal cycle: PREIN, A. F., LANGHANS, W., FOSSER, G., FERRONE, A., BAN, N., GOERGEN, K., KELLER, M., TOELLE, M., GUTJAHR, O., FESER, F., BRISSON, E., KOLLET, S., SCHMIDLI, J., VAN LIPZIG, N. P. M., LEUNG, R., 2015: A review on regional convection-permitting climate modeling: Demonstrations, prospects, and challenges. Reviews of Geophysics 53, (Heft 2), S. 323–361.

In general the authors seem not be very familiar with articles written in Europe.

5. Figures 5-11 Please add to each of the figures an additional one showing the absolute values of the observations. Otherwise it is hard to judge whether biases are large or small.

6. Figure 10 The authors write in the text (page 10, line 30) "... LRM simulation exhibits statistically significant positive temperature (up to 3 K) biases throughout the area (see

Fig. 10a), ..." (meaning biases to ERA-Interim). From the the biases of HRM and RRM are then even larger compared to ERA-Interim (Fig. 10b up to 6 K and the same in Fig. 10c, but for a smaller region). These are quite large biases. Can the authors give any explanation why the temperature bias doubles for the high-resolution model versions?

**# Technical Corrections**

1. Figure 15 Please enlarge the legend circle at the top right and add an additional radius axis with precipitation values.

---

## Referee Comment (RC2) · Anonymous Referee #2 · 16 Apr 2019

The authors of the reviewed manuscript present a framework for testing physics parametrizations on there scale awareness. They analyse the differences in a globally refined model (RRM) against a global high-resolution model (HRM). The advantage of a refined model lies in the computational cost. The authors state that the RRM model can mimic the outcome of the HRM model and can be used for future developments. Here, a selected area over the United States (CONUS) is used for the model evaluation. The authors compare the model results, both of HRM and RRM against measurements. The focus is set on hydrological values as precipitation. In addition, the authors focus on variables suitable for dynamical characterisation.

The manuscript needs major revisions. First of all, the category of this paper

should be revised. As stated on the web page for manuscript types: https://www.geoscientific-model-development.net/about/manuscript_types.html#item2 Development and technical papers should include details about the model improvements in terms of speed or accuracy. Here, some additional benchmarks in terms of computational time are missing. In addition, while reading the manuscript, I got the overall impression, that the authors are focussing more on the used parameterizations, than the code development. This focus has to be shifted, in order to address the demands of this specific manuscript type.

The overall objective is also a bit vague. Based on the abstract only, I can get three objectives. These are climate applications, usage as a testbed for physics parametrisations and hydrologic research. These are three points only discussed in a shallow way. I suggest to aim for the point of testbed usage. Here, the side by side comparisons shown in the result section would gain weight and the authors could discuss a bit more how to tackle occurring obstacles in the RRM model against HRM.

The introduction is a bit packed with acronyms. It is a overwhelming while reading the manuscript for the first time. In general, I would recommend a table of acronyms in the end. Plus, it would end in a better readability if less acronyms would be used.

The results you show are limited do DJF and JJA. I suggest to provide other seasons as well and at least discuss why you are not using these seasons for more intense studies. The last general point is, that I would wish for a detailed description of the technical implementation of the RRM.

After the major revision, the manuscript may be published in GMD.

**Specific comments:**

1. P1L13 *[. . . ] process-level representation at finer resolutions is a pressing need in order [. . . ].*
Could you please add a number here? What means finer? Finer than . . . ?

2. P1L14 *[. . . ] regarding extreme events in a changing climate.*

[Figure]

This is a bit too vague. Why do model simulations in a higher resolution provide more information to policy makers? I would suggest to rewrite the start of the abstract to get a bit more precise.

3. P1L22/23 *Differences between the RRM and HRM over the HR region are predominantly small, demonstrating that the RRM reproduces both well- and poorly simulated behaviours of the HRM over the CONUS.*
Again, too vague. In addition, the phrasing is a bit misleading. Roughly speaking, you have the same errors in both models. Please, rephrase.

4. P1L25 *influenced by the different parameter choices[...]*
This is a statement that is very obvoius. I won't mention that in an abstract.

5. P1L31 *[...] hydrologic research.*
No given context.

6. P2L4 *finer in the horizontal*
Finer means xx degrees?

7.P2L6 *and they must often be done at HR.*
I suggest to remove that part of the sentence.

8.P2L8 *one-year HR (0.25 average)*
Already defined - please erase

9.P2L12-15 *The RRM simulation cost is usually dominated by the computational cost of the HR region, and thus the total model cost is roughly proportional to the size of the region with finer resolution, referred to as a "mesh" (typically chosen to be about 10% of the globe, making the cost about 10% of a uniform HRM simulation.*
This is a bit self explanatory. Could you please include more details about the load balancing, computational costs, scaling factors and memory efficiency?

10. P2L16 *E3SMv2 science goals*
Please explain.

P2L20 *with 3 km horizontal grid*
Please be consistent with the units here. You've started with degrees, now switching to km.

P3L5 *play in future E3SM scientific and energy applications*
The aspect of energy application just pop up right here. Could you please describe those in more detail?

P3L12 *when increasing model resolution.*
You mean the horizontal resolution?

P3L30-32

*related to fast physics [. . . ] [. . . ] to fast physics, such as clouds and precipitation.*

Please add a careful definition of fast and slow physics here.

P4L15-20 This paragraph is much too long. The phrasing of e.g. "higher" should be more precises, e.g. higher than . . .

P4L26 Here, you name the definition of LRM and HRM again. Just leave that out.

P4L29 *[. . . ] tested over the Tropical Western Pacific (TWP) and the Eastern North Atlantic (ENA)*
Why is that worth mentioning?

P5L19 *[. . . ] associated with fast atmospheric physical processes.*
As mentioned above, a careful definition of fast and slow physics would be good.

P5L31 I'm not fully convinced here, that 5 years of simulation is really an appropriate time frame in terms of statistical significance. You also started your manuscript pin pointing to the climate aspect. While doing five years of simulation, the climate aspect is not covered what so ever. If you want to go more in the direction of the climate simulation, there is a need for a better discussion either why five years are really sufficient or perform more simulations. Thinking about ensemble, timeslice or just longer time periods.

P6L1 No need for the wording "skilful" here.

P6L18 How did you do this selection? Please describe the criteria you've chosen.

P6L24 *model results are conservatively interpolated*
What do you mean by *conservatively*? Please add more details on the interpolation method.

P6L30 This sentence is way too self explanatory.

P7L10 Why does the land surface model behave so differently? Please comment on that and add more details to that finding.

P7L33 Please elaborate on the point of improved physical processes.

P10L16-21 I do miss numbers in this paragraph. Please provide additional information if you use the phrase *too strong* or *relatively small*.

P12L13 I'm missing additional information about the interpolation method.

P13L8 Do you have any strategies of tackling that problem, that the model is not able to represent the night time maximum?

P13L29 *other models* Any references? What other models do you mean?

P13L31 *efficient tool for parameterization testing* This is an interesting point. I'm strongly encouraging you in working on that aspect a bit more and provide more details.

P15L11 Do you really mean many, or just those two?

P15L25 I disagree that the only conclusion is to develop better scale aware parameterizations. There are also model developments were you end up having e.g. convection permitting simulations.
P15L28 I'm a bit confused about the phrase *detailed guidance*, could you please comment on that?

**Technical comments**

P1L12 Climate simulation => Plural please

P3L2 all => All Make it two sentences

P5L7 analysed vs. analyzed

P5L26 nudging => nudged

P5L31 considered *as* spin-up

P11L32 *overwhelming* is not a proper term here.

P14L32 analysed vs. analyzed

P20L18 Please check you bibliography => N/A N/A

P22L14 Please add a date of last access.

Figure 1, 12 and 13 should be enlarged.
* * *

---

## Author Comment (AC1) · 15 May 2019

We thank the two reviewers for their time and effort in the review process. We appreciate their helpful and constructive comments and suggestions, which improve the quality of the paper. We have taken into account all the comments and revised our manuscript accordingly. Please see below for details (Reviewer comments are in blue).

**Anonymous Referee #1**

**#** General Comments**

In the present manuscript the authors analyse whether a globally refined model (RRM) can be used instead of a globally high-resolution model (HRM). The RRM is computationally less demanding than the HRM. The authors compare different atmospheric quantities of simulations with the RRM, the HRM and a coarse grid global model (LRM). The RRM was run in to different configurations. One using the LRM physics and the other the HRM physics. The authors conclude that the RRM is a useful tool for high resolution developments.

The manuscript needs major additions and major revisons (see specific comments below). After taking care of these the manuscript may be published in GMD.

**#** Specific Comments**

**Comments regarding the GMD principal criteria**

**Scientific significance - Excellent**

I am not aware of any publication that analyses in detail the effect of coarse grid global model, fine grid global model, coarse grid model with regional refined grid and tests of different physics tuning in the regional refinement.

Thanks. We are glad that our study fills this gap in the literature.

**### Scientific quality - Good**

The overall scientific quality is good. A major deficiency is that the cited literature in the manuscript is heavily "American" weighted. The authors should increase their literature study of articles outside of the U.S.. They would have been aware of the remarks under item 4 in "Other specific comments" below.

Thanks for pointing out this limitation. We revised the manuscript to include more non-US references as suggested. See the response to specific comments below for details.

**### Scientific reproducibility - Poor**

The link to the source code of the model is given at the end of the manuscript https://github.com/E3SM-Project/E3SM. However, this information is not enough to reproduce the science by a fellow scientist. What at least is missing:

- exact information where to get the observational and reanalysis data from. This can be provided by web links in table 2. Even this may be not enough, therefore comments on what has to be taken into account additionally should be added if needed. The web links of the observational and reanalysis data used in this study were added in the last column of Table 2. These locations are also documented in the acknowledgement session.

- detailed technical model simulation setup (can be put into the Appendix). Actually, every information that is needed to run the model simulations as the authors did.

We added two new Tables (Tables A1 and A2) in the Appendix for the model simulation setup details, including code hash numbers, grids, configurations, and non-default parameter changes.

**- post processing tools used for the statistical analyses**

We used the NCL built-in function "ttest" for the Student's t-test in the paper.

- time period(s) of simulations (see item 3 under "Other specific comments" below.

See answers in item 3 below.

**presentation quality - Good**

The presentation of the results is overall good in a well-structured way. However, it misses some important information (see "Other specific comments" below)

**Other specific comments**

1. The authors claim to use a "fully-coupled Earth system model", however, in "2.1 Model overview and experiment design" they describe only the atmpospheric part. What about the ocean? What about the coupling?

Sorry for the confusion. The fully-coupled Earth system model (E3SM) shares the same code repository with its atmosphere model (EAM). Therefore, with the same code repository, users have the option to run either the coupled simulation or the atmosphere-only simulation. We only used the atmosphere-only simulations in the present study, and thus only described the atmosphere model.

We added the following clarification on P4 L10, "Since all the simulations analysed here are atmosphere-only ones, we only provide information about the atmosphere model. Details about the coupled E3SM model can be found in Golaz et al. (2019).".

2. page 5, line 7-8 "The LRM and HRM physics time steps are 30 minutes and 15 minutes, respectively". What about the dynamic time step? The dynamic time step depends on the grid width. How do the authors set the time step in the coarse grid region of the RRM and that one in the high-resolution domain?

The dynamics use 3 layers of substepping. For the LRM (HRM), the Lagrangian vertical discretization timestep is 15 minutes (2.5 minutes), the horizontal discretization timestep is 5

minutes (75 seconds), and the explicit numerical diffusion timestep is 100 seconds (18.75 seconds). In RRM, the same time steps are used for both the low- and high-resolution domains. The RRM\_LR simulation uses the LRM time steps, while other RRM simulations use the HRM steps. We have revised the manuscript to include this dynamic timestep information on page 5.

3. Page 5, line 30 -34 "All free-running simulations are run for a period of 5 years. The first year is considered spin-up, thus we study the results from the last 4 years. The nudging run simulates year 2011, whereas the AMIP results are extracted for year 2011 from a long simulation starting from 1870 (Golaz et al., 2019). Model output is stored as monthly and hourly averages." This is the only part in the manuscript where I can find an information on the time period. From Table 2 it seems that all observations have different time periods. Are the time periods listed there are also the time periods which are compared with the model results?

This is correct. In this study, we mainly focus on the modeled climatologies. The time periods listed in Table 2 are used to calculate the observed climatologies to compare with the model results.

4. Page 13, line 9-11 "... incorrect diurnal cycles. The similarity between RRM and HRM indicates that RRM simulations will be valuable for understanding and addressing this important model bias." and line 28-31 "... the time of peak precipitation is a few hours early, consistent with the experience of other models. More physically based improvements are needed to find a solution to the summertime diurnal cycle issue for precipitation over the CONUS, and the RRM provides an efficient tool for parameterization testing."

As the authors state right this is a well-known problem of coarse grid models. However, they did not cite any papers on possible solutions.

A possible solution for the ECMWF convection scheme is described in BECHTOLD, P., SEMANE, N., LOPEZ, P., CHABOUREAU, J.-P., BELJAARS, A., BORMANN, N., 2014: Representing Equilibrium and Nonequilibrium Convection in Large-Scale Models. Journal of the Atmospheric Sciences 71, (Heft 2), S. 734–753.

In general this problem does not appear in convection permitting simulations ( < about 2-4 km grid width). A starting point for reading on the effect of convective permitting simulations on the diurnal cycle: PREIN, A. F., LANGHANS, W., FOSSER, G., FERRONE, A., BAN, N., GOERGEN, K., KELLER, M., TOELLE, M., GUTJAHR, O., FESER, F., BRISSON, E., KOLLET, S., SCHMIDLI, J., VAN LIPZIG, N. P. M., LEUNG, R., 2015: A review on regional convection-permitting climate modeling: Demonstrations, prospects, and challenges. Reviews of Geophysics 53, (Heft 2), S. 323–361. In general the authors seem not be very familiar with articles written in Europe.

Thanks for pointing out this important information we missed in our manuscript. At the end of Section 3, we added the following text and references.

Previous studies (e.g., Bechtold et al., 2004; Stratton and Stirling, 2012; Bechtold et al., 2014) provide possible solutions for this issue of simulating the diurnal cycle of convective precipitation over land by modifying convective trigger procedures, entrainment, and convective closures. Our recent study (Xie et al., 2019) shows substantial improvement in the

precipitation diurnal cycle in the LRM by employing a new convective trigger with a dynamic constraint on the convection onset, and with the capability of detecting moist instability above the boundary layer. We will apply the RRM testbed to extend the new convective trigger to the HRM and report the results in a future paper. This bias in the diurnal cycle of convection is significantly improved in convection-permitting (horizontal grid spacing < 2-4 km) simulations (Prein et al., 2015). The E3SM project is making progress in developing its convection-permitting version (E3SMv4), for which the RRM testbed will be heavily relied on.

**5. Figures 5-11 Please add to each of the figures an additional one showing the absolute values of the observations. Otherwise it is hard to judge whether biases are large or small.**

We added the observation plots as panel (e) on these figures. We also reversed the colormap for the difference plots of precipitation and total precipitable water – blue represents wetter, while brown drier.

6. Figure 10 The authors write in the text (page 10, line 30) "... LRM simulation exhibits statistically significant positive temperature (up to 3 K) biases throughout the area (see Fig. 10a), ..." (meaning biases to ERA-Interim). From the the biases of HRM and RRM are then even larger compared to ERA-Interim (Fig. 10b up to 6 K and the same in Fig. 10c, but for a smaller region). These are quite large biases. Can the authors give any explanation why the temperature bias doubles for the high-resolution model versions?

The warm bias can be roughly attributed to two separate sources: the evaporative fraction (EF) contribution and the radiation contribution which is primarily caused by excessive absorbed solar radiation at the surface. EF is defined as the fraction of the combined latent and sensible heat fluxes that are in latent form. Models with too low EF tend to use the radiative input to heat the surface instead of evaporating water (Ma et al., 2018).

We applied the diagnostic (Equation 1 of Ma et al., 2018) to our model results and found that the larger temperature bias in the high-resolution model versions is because the EF contribution is a few times larger with enhanced resolution, while the radiation contribution remains almost unchanged.

We added the following in this paragraph. "This warm bias can be roughly attributed to two separate sources (Ma et al., 2018): the evaporative fraction (EF) contribution and the radiation contribution which is primarily caused by excessive absorbed solar radiation at the surface. EF is defined as the fraction of the combined latent and sensible heat fluxes that are in latent form. Models with too low EF tend to use the radiative input to heat the surface instead of evaporating water. The larger bias in the HRM is because the EF contribution is a few times larger with enhanced resolution, while the radiation contribution remains almost unchanged."

**# Technical Corrections**

1. Figure 15 Please enlarge the legend circle at the top right and add an additional radius axis with precipitation values.

We revised Figure 15 with the enlarged legend circle. Now the legend text and the radius axis inside the circle are shown clearer.

**Anonymous Referee #2**

The authors of the reviewed manuscript present a framework for testing physics parametrizations on there scale awareness. They analyse the differences in a globally refined model (RRM) against a global high-resolution model (HRM). The advantage of a refined model lies in the computational cost. The authors state that the RRM model can mimic the outcome of the HRM model and can be used for future developments. Here, a selected area over the United States (CONUS) is used for the model evaluation. The authors compare the model results, both of HRM and RRM against measurements. The focus is set on hydrological values as precipitation. In addition, the authors focus on variables suitable for dynamical characterisation.

The manuscript needs major revisions. First of all, the category of this paper should be revised. As stated on the web page for manuscript types: https://

www.geoscientific-model-development.net/about/manuscript\_types.html#item2 Development and technical papers should include details about the model improvements in terms of speed or accuracy. Here, some additional benchmarks in terms of computational time are missing. In addition, while reading the manuscript, I got the overall impression, that the authors are focussing more on the used parameterizations, than the code development. This focus has to be shifted, in order to address the demands of this specific manuscript type.

We agree that the manuscript needs to be revised to better fit the "development and technical" category. We made the following major changes:

- Added the model speed and cost information in Table 1 to benchmark the RRM computational performance and highlight its computational efficiency relative to the uniform high-resolution model.
- Revised the title to "Regionally refined testbed in E3SM Atmosphere Model Version 1 (EAMv1) and applications for high-resolution modelling" to emphasize the RRM technical usage as a physics parameterization testbed for high-resolution modelling.
- We included more information about the code development of implementing the RRM.

The overall objective is also a bit vague. Based on the abstract only, I can get three objectives. These are climate applications, usage as a testbed for physics parametrisations and hydrologic research. These are three points only discussed in a shallow way. I suggest to aim for the point of testbed usage. Here, the side by side comparisons shown in the result section would gain weight and the authors could discuss a bit more how to tackle occurring obstacles in the RRM model against HRM. We tried to state the overall goal of this paper as clearly as possible, but we agree it can be improved. We like the suggestion of aiming for the testbed usage. As mentioned above, the title was changed to explicitly point out RRM as a testbed. We also revised the whole text to focus on the testbed point. For instance, the last sentence of the abstract was revised to "...the RRM is an efficient and effective testbed for HRM development", and the last sentence of the first paragraph of Section 2.1 was changed to "...we focus on the EAMv1 regionally refined testbed capability over the CONUS domain".

The introduction is a bit packed with acronyms. It is a overwhelming while reading the manuscript for the first time. In general, I would recommend a table of acronyms in the end. Plus, it would end in a better readability if less acronyms would be used. The results you show are limited do DJF and JJA. I suggest to provide other seasons as well and at least discuss why you are not using these seasons for more intense studies. The last general point is, that I would wish for a detailed description of the technical implementation of the RRM.

Thanks for the suggestion. We included the acronym table (Table 4) in the revised version as suggested. Regarding the description of the technical implementation of the RRM, we added the following in the second paragraph of Section 2.1, "We created the regionally refined grid with the offline software tool, Spherical Quadrilateral Grid Generator (SQuadGen, https://github.com/ClimateGlobalChange/squadgen)." and "We used a new tensor hyperviscosity formulation (Guba et al., 2014) to eliminate numerical noise and oscillations."

Our main purpose here is to document the RRM testbed with "proof-of-concept" examples instead of an intense study. The DJF and JJA are the two extreme seasons, while MAM and SON are more moderate with the results somewhat in between DJF and JJA. Therefore, we chose to only show DJF and JJA. We added this reason at the beginning of the results section (Section 3) as "In this section, we will focus on the results of June-July-August (JJA) and December-January-February (DJF), the two more extreme seasons at the CONUS in a year when some long-standing systematic model errors are present.".

After the major revision, the manuscript may be published in GMD. Specific comments: 1. P1L13 [...] process-level representation at finer resolutions is a pressing need in order [...]. Could you please add a number here? What means finer? Finer than ...?

We added "< 100 km" in the sentence. The specific resolution requirements vary by the extreme events that the policies target. 100 km is a typical resolution used in the current climate simulations and is often too coarse.

2. P1L14 [...] regarding extreme events in a changing climate. This is a bit too vague. Why do model simulations in a higher resolution provide more information to policy makers? I would suggest to rewrite the start of the abstract to get a bit more precise.

We revised the sentence to "... in order to provide more detailed actionable information...". Extreme events often occur at high spatial and temporal scales. High-resolution simulations can provide more details about these events than can the low-resolution simulations.

3. P1L22/23 Differences between the RRM and HRM over the HR region are predominantly small, demonstrating that the RRM reproduces both well- and poorly simulated behaviours of the HRM over the CONUS.

Again, too vague. In addition, the phrasing is a bit misleading. Roughly speaking, you have the same errors in both models. Please, rephrase.

Thank you, we have adjusted the text to be clearer.

4. P1L25 *influenced by the different parameter choices*[. . . ] This is a statement that is very obvoius. I won't mention that in an abstract.

Considering one of the most important applications of RRM as a testbed is to test the model performance with different parameters, we prefer to keep this sentence.

5. P1L31 [. . . ] hydrologic research. No given context.

Removed. Thanks.

6. P2L4 *finer in the horizontal* Finer means xx degrees?

At the end of this paragraph, we mentioned that the E3SM plans to develop global cloudresolving model with 3 km resolution. Finer means the resolution between 3 km and 25 km.

7.P2L6 and they must often be done at HR. I suggest to remove that part of the sentence.

Deleted.

8.P2L8 *one-year HR (0.25 average)* Already defined - please erase

We changed to "one-year 0.25° HR". It is worth mentioning the resolution here to avoid confusion.

9.P2L12-15 The RRM simulation cost is usually dominated by the computational cost of the HR region, and thus the total model cost is roughly proportional to the size of the region with finer resolution, referred to as a "mesh" (typically chosen to be about 10% of the globe, making the cost about 10% of a uniform HRM simulation.

This is a bit self explanatory. Could you please include more details about the load balancing, computational costs, scaling factors and memory efficiency?

Thank you for the suggestions. We feel that this sentence is probably useful for the audience who are less familiar with the RRM to gain a rough idea about how much RRM can save relative to the uniform high-res model. We prefer to keep it.

We added details about the speed (simulation years per day), node numbers, and the cost (core-hours per year) of all the model settings used in this study in the revised Table 1 to give more information about the technical aspects of the model. It shows that in the simulations analyzed here the RRM cost is 13%-15% of the uniform high-res model, depending on the parameter choices.

**10. P2L16 *E3SMv2 science goals* Please explain.**

The manuscript has been revised to include more details: "...E3SMv2 science goal of understanding the relative impacts of global forcing versus regional influences of human activities on flood and drought in North America."

11 P2L20 *with 3 km horizontal grid* Please be consistent with the units here. You've started with degrees, now switching to km.

Traditionally, degrees are used for coarse resolution global models and km for cloud-resolving models. We switched the units to be consistent with this convention.

12 P3L5 *play in future E3SM scientific and energy applications* The aspect of energy application just pop up right here. Could you please describe those in more detail?

We changed the sentence to "...in future E3SM scientific applications".

13 P3L12 *when increasing model resolution.* You mean the horizontal resolution?

We meant both the horizontal and vertical resolutions. Changed to "when increasing model horizontal or vertical resolution."

14 P3L30-32 related to fast physics [...] [...] to fast physics, such as clouds and precipitation. Please add a careful definition of fast and slow physics here.

We added the definitions of fast vs. slow physics by revising the text to "...between short (a few days) and long (seasonal to annual) timescale systematic errors in climate models...".

15 P4L15-20 This paragraph is much too long. The phrasing of e.g. "higher" should be more precises, e.g. higher than . . .

We changed the text to read "a higher (~0.1 hPa compared to 2 hPa) model top".

16 P4L26 Here, you name the definition of LRM and HRM again. Just leave that out.

Done. Thanks.

17 P4L29 [...] tested over the Tropical Western Pacific (TWP) and the Eastern North Atlantic (ENA) Why is that worth mentioning?

This paper serves as the main documentation of the EAMv1 RRM, so we think it is worth pointing out the available RRM grids over other regions in EAMv1.

18 P5L19 [...] *associated with fast atmospheric physical processes.* As mentioned above, a careful definition of fast and slow physics would be good.

Agreed. We defined the timescales of fast and slow physics as suggested (see above).

19 P5L31 I'm not fully convinced here, that 5 years of simulation is really an appropriate time frame in terms of statistical significance. You also started your manuscript pin pointing to the climate aspect. While doing five years of simulation, the climate aspect is not covered what so ever. If you want to go more in the direction of the climate simulation, there is a need for a better discussion either why five years are really sufficient or perform more simulations. Thinking about ensemble, timeslice or just longer time periods.

The reasons why 5 years simulations are appropriate are because our present study focuses on hydrological variables, for example precipitation, and we mainly use the atmosphere-only simulations with cycling climatologically forcings and emissions.

Previous studies (Xie et al., 2012; Ma et al., 2014) have demonstrated a strong correspondence between short and long timescale systematic errors in climate models for fields related to fast physics, such as precipitation and clouds. We verified this finding with a 20-year LRM simulation and found that the 5-year climatology and 20-year climatology of the variables analyzed here are very similar. Also considering the expensive cost of multi-decadal HRM simulations, we opted to use 5-year simulations for all resolutions.

**20 P6L1 No need for the wording "skilful" here.**

We want to emphasize the link between better model skills and more realistic simulation. We intend to keep it.

**21 P6L18 How did you do this selection? Please describe the criteria you've chosen.**

We mainly follow the Program for Climate Model Diagnosis and Intercomparison (PCMDI) Metrics Package (PMP). We added the citation to Gleckler et al. (2016).

**22 P6L24 model results are conservatively interpolated What do you mean by conservatively? Please add more details on the interpolation method.**

We used the "conserve" method of the Earth System Modeling Framework (ESMF) regridding software. The text was revised to "model results are conservatively interpolated (with the "conserve" method of the Earth System Modeling Framework (ESMF, https://www.earthsystemcog.org/projects/esmf/) regridding software) to...".

**23 P6L30 This sentence is way too self explanatory.**

That is true, but we received feedback from the readers that they somehow thought the goal of our analysis is to show the RRM outperforms the HRM. We, therefore, decided to make it clear.

**24 P7L10 Why does the land surface model behave so differently? Please comment on that and add more details to that finding.**

This is mainly due to the different surface energy partitioning in the land surface models. The fraction of sensible heat flux is much higher in the HRM than in the LRM. Please see the response to question 6 of the specific comments of reviewer #1 for details. We changed the sentence to "...along with feedbacks (surface energy partitioning shifting towards more sensible heat flux) from the land surface model.".

25 P7L33 Please elaborate on the point of improved physical processes.

Changed to "...improved representation of physical processes by better physical parameterizations."

26 P10L16-21 I do miss numbers in this paragraph. Please provide additional information if you use the phrase too strong or relatively small.

We included the numbers where strong or small are used in this paragraph.

27 P12L13 I'm missing additional information about the interpolation method.

The interpolation method was provided as "...are interpolated with the ESMF conservative regridding method to...".

**28 P13L8 Do you have any strategies of tackling that problem, that the model is not able to represent the night time maximum?**

A new convective trigger can help reduce this night time precipitation bias. The new trigger uses a dynamic constraint on the convection onset and has the capability of detecting moist instability above the boundary layer. The details about this recent study was reported by Xie et al., (2019).

**29 P13L29 other models Any references? What other models do you mean?**

We added the references to Dai et al., 1999; Stratton and Stirling, 2012; Bechtold et al., 2014 for the NCAR regional climate model, Met Office Hadley Centre climate model, and the ECMWF model.

**30 P13L31 efficient tool for parameterization testing This is an interesting point. I'm strongly encouraging you in working on that aspect a bit more and provide more details.**

The RRM is much more efficient than the HRM in terms of its computational cost, throughput, and storage requirements. Our revised Table 1 includes the speed and cost details for RRM and HRM. It clearly shows that the RRM only costs 15% of the HRM with a 15% faster throughput and requires much less (13%) compute nodes.

**31 P15L11 Do you really mean many, or just those two?**

Deleted "many".

32 P15L25 I disagree that the only conclusion is to develop better scale aware parameterizations. There are also model developments were you end up having e.g. convection permitting simulations.

Revised to "...the need to develop better scale-aware physical parameterizations or convectionpermitting simulations in the future."

**33 P15L28 I'm a bit confused about the phrase detailed guidance, could you please comment on that?**

We included Table 3 with all the parameters for the non-US nudging simulation analyzed in this paper as an example for users who are interested in running EAMv1 RRM with nudging.

**Technical comments P1L12 Climate simulation => Plural please**

Corrected. Thanks.

P3L2 all => All Make it two sentences

Done.

P5L7 analysed vs. analyzed

GMD requires the British spelling "analysed".

P5L26 nudging => nudged

Done.

P5L31 considered as spin-up

Thank you. Done.

P11L32 overwhelming is not a proper term here.

Revised to "dominant".

P14L32 analysed vs. analyzed

See above.

P20L18 Please check you bibliography => N/A N/A

The page information was corrected.

P22L14 Please add a date of last access.

Done

Figure 1, 12 and 13 should be enlarged.

We changed Figure 1. We will enlarge Figures 12 and 13 during the typesetting process.

**Reference:**

Bechtold, P., Chaboureau, J.-P., Beljaars, A., Betts, A. K., Köhler, M., Miller, M. and Redelsperger, J.-L.: The simulation of the diurnal cycle of convective precipitation over land in a global model, Q. J. R. Meteorol. Soc., 130(604), 3119–3137, doi:10.1256/qj.03.103, 2004.

Bechtold, P., Semane, N., Lopez, P., Chaboureau, J.-P., Beljaars, A. and Bormann, N.: Representing Equilibrium and Nonequilibrium Convection in Large-Scale Models, J. Atmospheric Sci., 71(2), 734–753, doi:10.1175/JAS-D-13-0163.1, 2014.

Ma, H. -Y., Klein, S. A., Xie, S., Zhang, C., Tang, S., Tang, Q., et al. (2018). CAUSES: On the Role of Surface Energy Budget Errors to the Warm Surface Air Temperature Error Over the Central United States. *Journal of Geophysical Research: Atmospheres*, *123*(5), 2888–2909. https://doi.org/10.1002/2017JD027194

Ma, H.-Y., Xie, S., Klein, S. A., Williams, K. D., Boyle, J. S., Bony, S., Douville, H., Fermepin, S., Medeiros, B., Tyteca, S., Watanabe, M. and Williamson, D.: On the Correspondence between Mean Forecast Errors and Climate Errors in CMIP5 Models, J. Clim., 27(4), 1781–1798, doi:10.1175/JCLI-D-13-00474.1, 2014.

Gleckler, P., Doutriaux, C., Durack, P., Taylor, K. E., Zhang, Y., Williams, D., Mason, E. and Servonnat, J.: A more powerful reality test for climate models, Eos Trans. Am. Geophys. Union, 97, doi:10.1029/2016E0051663, 2016.

Guba, O., Taylor, M. A., Ullrich, P. A., Overfelt, J. R. and Levy, M. N.: The spectral element method (SEM) on variable-resolution grids: evaluating grid sensitivity and resolution-aware numerical viscosity, Geosci. Model Dev., 7(6), 2803–2816, doi:https://doi.org/10.5194/gmd-7-2803-2014, 2014.

Prein, A. F., Langhans, W., Fosser, G., Ferrone, A., Ban, N., Goergen, K., Keller, M., Tölle, M., Gutjahr, O., Feser, F., Brisson, E., Kollet, S., Schmidli, J., Lipzig, N. P. M. van and Leung, R.: A review on regional convectionpermitting climate modeling: Demonstrations, prospects, and challenges, Rev. Geophys., 53(2), 323–361, doi:10.1002/2014RG000475, 2015.

Stratton, R. A. and Stirling, A. J.: Improving the diurnal cycle of convection in GCMs, Q. J. R. Meteorol. Soc., 138(666), 1121–1134, doi:10.1002/qj.991, 2012.

Xie, S., Ma, H.-Y., Boyle, J. S., Klein, S. A. and Zhang, Y.: On the Correspondence between Short- and Long-Time-Scale Systematic Errors in CAM4/CAM5 for the Year of Tropical Convection, J. Clim., 25(22), 7937–7955, doi:10.1175/JCLI-D-12-00134.1, 2012.

Xie, S., Wang, Y.-C., Lin, W., Ma, H.-Y., Tang, Q., Tang, S., Zheng, X., Golaz, J.-C., Zhang, G. and Zhang, M.: Diurnal Cycle of Precipitation Simulated by E3SM with a Revised Convective Trigger, J. Adv. Model. Earth Syst., submitted, 2019.